# NEW INSIGHTS ON REDUCING ABRUPT REPRESENTATION CHANGE IN ONLINE CONTINUAL LEARNING

**Lucas Caccia**[*]
McGill University, Mila
Facebook AI Research

**Rahaf Aljundi**
Toyota Motor Europe

**Nader Asadi**
Concordia University, Mila

**Tinne Tuytelaars**
KU Leuven

**Joelle Pineau**
McGill University, Mila
Facebook AI Research

**Eugene Belilovsky**
Concordia University, Mila

## ABSTRACT

In the online continual learning paradigm, agents must learn from a changing distribution while respecting memory and compute constraints. Experience Replay (ER), where a small subset of past data is stored and replayed alongside new data, has emerged as a simple and effective learning strategy. In this work, we focus on the change in representations of observed data that arises when previously unobserved classes appear in the incoming data stream, and new classes must be distinguished from previous ones. We shed new light on this question by showing that applying ER causes the newly added classes' representations to overlap significantly with the previous classes, leading to highly disruptive parameter updates. Based on this empirical analysis, we propose a new method which mitigates this issue by shielding the learned representations from drastic adaptation to accommodate new classes. We show that using an *asymmetric* update rule pushes new classes to adapt to the older ones (rather than the reverse), which is more effective especially at task boundaries, where much of the forgetting typically occurs. Empirical results show significant gains over strong baselines on standard continual learning benchmarks [1].

## 1 INTRODUCTION

Continual learning is concerned with building models that can learn and accumulate knowledge and skills over time. A continual learner receives training data sequentially, from a potentially changing distribution, over the course of its learning process. The distribution change might be either a shift in the input domain or new categories being learned. The main challenge is to design models that can learn how to use the new data and acquire new knowledge, while preserving or improving the performance on previously learned data. While different settings have been investigated of how new data are being received and learned, we focus on the challenging scenario of learning from an *online* stream of data with *new classes being introduced at unknown points in time* and where *memory and compute constraints* are applied on the learner. Additionally, we assume a shared output layer among all the learned classes (Aljundi et al., 2019b). This setting is different and harder than the conventional multi-head setting (Farquhar & Gal, 2018) where each new group of classes is considered as a new task with a dedicated head (classification layer), requiring a task oracle at test time to activate the correct head. The axes of our setting (online learning, no task boundary, no test time oracle, constant memory, and bounded compute) align with the main desiderata of continual learning as described in De Lange et al. (2019).

Catastrophic forgetting (McCloskey & Cohen, 1989), where previous knowledge is overwritten as new concepts are learned, remains a key challenge in the online continual learning setting. To prevent forgetting, methods usually rely on storing a small buffer of previous training data and replaying samples from it as new data is learned. This can partially counteract catastrophic forgetting, but

---

[*]Corresponding Author `lucas.page-caccia@mail.mcgill.ca`

[1]Code to reproduce experiments is available at `www.github.com/pclucas14/AML`

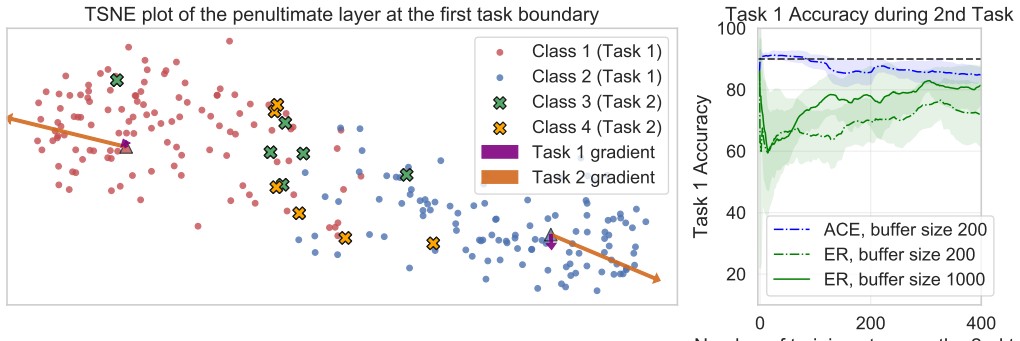

Figure 1: (left) Analysis of representations with the first task's class prototypes *at a task boundary*. Under ER when Task 2 begins, class 1 & 2 prototypes experience a large gradient and subsequent displacement caused by the close location of the unobserved sample representations, this leads to a significant drop in performance (right). Our proposed method (ACE) mitigates the representation drift issue and observes no performance decrease on a task switch.

still tends to lead to large disruptions in accuracy, particularly at the initial task boundary or shift in distribution. Various works focus on studying which samples to store (Borsos et al., 2020; Aljundi et al., 2019b) or which samples to replay when receiving new data (Aljundi et al., 2019a). In this work, we direct our attention to the representations being learned and investigate how the features of previously learned classes change and drift over time.

Consider the time point in a stream when a new class is introduced after previous classes have been well learned. If we consider the representation being learned, incoming samples from new classes are likely to be dispersed, potentially near and between representations of previous classes, while the representations of previous classes will typically cluster according to their class. Indeed, one might expect minimal changes to the learned representation of the previous classes, while the new classes samples are pushed away from the clusters of old class data. However, with a standard Experience Replay (ER) algorithm (Chaudhry et al., 2019), we observe that it is the representations of older classes that is heavily perturbed after just a few update steps when training on the new class samples. We hypothesize that the fundamental issue arises from the combination of: new class samples representations lying close to older classes and the loss structure of the standard cross entropy applied on a mix of seen and unseen classes. We illustrate the observed effect in Fig. 1 (left).

This behavior is exacerbated especially in the regime of low buffer size. With larger replay buffers, the learner can recover knowledge about the prior classes over time, while with smaller buffers the initial disruptive changes in representations are challenging to correct. Indeed we illustrate this effect in Fig. 1 (right), we see that ER only recovers from the initial displacement given a much larger buffer size.

In standard continual learning with replay (Aljundi et al., 2019a; Chaudhry et al., 2019) the same loss function is usually employed on both the newly received samples and the replayed samples. In contrast, we propose a simple and efficient solution to mitigate this representation drift by using **separate losses on the incoming stream and buffered data**. The key idea is to allow the representations of samples from new classes to be learned in isolation of the older ones first, by excluding the previously learned classes from the incoming data loss. The discrimination between the new classes and the older ones is learned through the replayed batches, but only after incoming data has been learned, added to the buffer, and made available for replay. To allow more direct control of the structure in representations we first consider a metric learning based loss for the incoming data, proposed in Khosla et al. (2020), where we propose to exclude samples of previously learned classes from the negative samples. We show that this type of negative selection is critical, and in contrast issues arise when negative examples are sampled uniformly from the buffer. These issues mimic those seen with standard losses in experience replay (ER) (Aljundi et al., 2019a). On the other hand we use a different loss on replay buffer data that is allowed to consider new and old classes, thereby consolidating knowledge across current and previous tasks. We call this overall approach ER with *asymmetric metric learning* (ER-AML).

Since cross entropy losses can be more efficient in training for classification than metric learning and contrastive losses (avoiding positive and negative selection) and it is widely used in incremental

and continual learning, we also propose an alternative cross entropy solution that similarly applies an asymmetric loss between incoming and replay data. Notably, the cross entropy applied to the incoming data *only considers logits of classes of the incoming data*. This variant, named ER with *asymmetric cross-entropy* (ER-ACE), along with ER-AML show strong performance, with little disruption at task boundaries Fig. 1 (right). We achieve state of the art results in existing benchmarks while beating different existing methods including the traditional ER solution with an average relative gain of 36% in accuracy. Our improvements are especially high in the small buffer regime. We also show that the mitigation of the old representation drift does not hinder the ability to learn and discriminate the new classes from the old ones. This property emerges from *only learning the incoming data in isolation*; as we will see, also isolating the rehearsal step (as in Ahn et al. (2020)) leads to poor knowledge acquisition on the current task. Furthermore we show our ER-ACE objective can be combined with existing methods, leading to additional gains. Finally, we take a closer look at the computation cost of various existing methods. We show that some methods, while obtaining good performance under standard evaluation protocols, fail to meet the computational constraints required in online CL. We provide an extensive evaluation of computational and memory costs across several baselines and metrics.

To summarize, our contributions are as follows. We first highlight the problem of representation drift in the online continual learning setting. We identify a root cause of this issue through an extensive empirical analysis (Sec. 4.2). Second, we propose a new family of methods addressing this issue by treating incoming and past data asymmetrically (Sec. 4.1, 4.3) . Finally, we show strong gains over replay baselines in a new evaluation framework designed to monitor real world constraints (Sec. 5). To the best or our knowledge, we are the first to report the computation costs of different methods in our setting, revealing new insights.

## 2 RELATED WORK

Research on continual learning can be divided based on the sequential setting being targeted (see Zeno et al. (2018); van de Ven & Tolias (2019); Normandin et al. (2021); Lesort et al. (2021) for categorizations of the settings and De Lange et al. (2019) for a broad survey on continual learning). Earlier works consider the relaxed setting of task incremental learning (Aljundi et al., 2017; Serrà et al., 2018; Li & Hoiem, 2016) where the data stream is divided into chunks of tasks and each task is learned offline with multiple iterations over the data of this task. While this setting is easier to handle as one task can be learned entirely, it limits the applicability of the solution.

In this work, we consider the challenging setting of an online stream of non-i.i.d. data where changes can anytime occur in the input domain or in the output space. This more realistic setting has attracted increasing interest lately (Lopez-Paz et al., 2017; Aljundi et al., 2019a). Specifically, we study the *single-head* (or shared head) setting, where when queried, the learner is not told which task the sample belongs to (as opposed to the *multi-head* setting). The single-head assumption is further studied in task-agnostic continual learning settings (He et al., 2019; Caccia et al., 2020; Ostapenko et al., 2021; Von Oswald et al., 2021) in which the task-boundary assumption, amongst others, is also relaxed. Many of the solutions to the online continual learning problem rely on the use of a buffer formed of previous memories which are replayed alongside new data during the learning process. Several works (Borsos et al., 2020; Chaudhry et al., 2019; Aljundi et al., 2019b) propose solutions to select which samples should be stored, or retrieved for replay (Aljundi et al., 2019a), or both (Shim et al., 2021). Lopez-Paz et al. (2017); Chaudhry et al. use replay to perform constrained optimization, limiting interference with previous tasks as new ones are learned. Our work, on the other hand, focuses on the appropriate loss function in this context. Tang & Matteson (2020) propose a graph-based approach that capture pairwise similarities between samples. Dark Experience Replay (DER) (Buzzega et al., 2020) suggests an alternative replay loss. Samples are stored along with their predicted logits and once replayed the current model is asked to keep its output close to the previously recorded logits. While the method is simple and effective it is worth noting that it relies heavily on data augmentation. Our work is orthogonal and can be combined with DER as we show in Sec. D. Finally, concurrent work (Mai et al., 2021) also use a contrastive loss for online continual learning, but not in an asymmetric fashion.

In our work we also investigate the underlying causes for performance degradation in replay-based methods. Related to this study are works in the class incremental setting, where similar to our case a shared output layer is used, *but classes are learned offline*. Works in this area address the implicit class imbalance issue occurring when new classes are learned alongside replayed data. Zhao et al.

(2019) proposes to correct last layer weights after a group of classes is learned via adjusting the weights norm. Wu et al. (2019) suggests to deploy extra additional parameters in order to linearly correct the "bias" in the shared output layer. Those parameters are learned at the end of each training phase. Hou et al. (2019) considers addressing this imbalance through applying cosine similarity based loss as opposed to the typical cross entropy loss along with a distillation loss and a margin based loss with negatives mining to preserve the feature of previous classes. Recently, Ahn et al. (2020) propose to learn the incoming tasks and the previous tasks separately. They use a masked softmax loss for the incoming and rehearsal data, to counter the class imbalance. All the methods highlighted above operate in the *offline* setting, where data from the current task can be revisited as needed making the disruptive issues emphasized at the task boundary less critical. In this paper, we focus on the online setting, with potentially overlapping tasks. As we will see, work by Ahn et al. (2020) developed to counter class imbalance, can inhibit learning of the current task in the online setting (see Appendix B). Lastly, Zeno et al. (2018) uses a logit masking related to our method but their context is based on the multi-head setting, and does not consider replay based methods, where learning across tasks occurs. Their goal is to activate only the head of which the samples within the new batch belong to. However, our approach is more general and it applies to the single head setting (where we have a single output layer for all classes, and no task oracle.)

## 3 LEARNING SETTING AND NOTATION

We consider the setting where a learner is faced with a possibly never-ending stream of data. At every time step, a labelled set of examples $(\mathbf{X}^{in}, \mathbf{Y}^{in})$ drawn from a distribution $D_t$ is received. However, the distribution $D_t$ itself is sampled at each timestep and can suddenly change to $D_{t+1}$, when a task switch occurs. The learner is not explicitly told when a task switch happens, nor can it leverage a task identifier during training or evaluation. We note that this definition generalizes task-incremental learning, where each task is seen one after the other. In this scenario, given $T$ tasks to learn, $D_t$ changes $T - 1$ times over the full steam, yielding $T$ locally i.i.d learning phases. We also explore in this paper a more general setting without the notion of clearly delineated tasks (Aljundi et al., 2018; Chen et al., 2020a), where the data distribution gradually changes over time.

Given a model $f_\theta(x)$ representing a neural network architecture with parameters $\theta$, we want to minimize the classification loss $\mathcal{L}$ on the newly arriving data batch while not negatively interfering with the previously learned classes (i.e. increasing the classification loss). A simple and efficient approach to achieve this is to replay stored samples from a fixed size memory, $\mathcal{M}$, in conjunction with the incoming data (Chaudhry et al., 2019; Rolnick et al., 2018). The core of our approach is that instead of treating the replayed batch and the incoming one similarly and naively minimizing the same loss, *we opt for a specific loss structure on the incoming batch that would limit the interference with the previously well learned classes*. We approach this by allowing the features of the newly received classes in the incoming data to be initially learned in isolation of the older classes. We first present our idea based on a metric learning loss and then generalize to the widely deployed cross-entropy loss.

## 4 METHODS

### 4.1 A DISTANCE METRIC LEARNING APPROACH FOR REDUCING DRIFT (ER-AML)

In order to allow fine-grained control of which samples will be pushed away from other samples given an incoming batch, we propose to apply, on the incoming data, a metric learning based loss from Khosla et al. (2020). Related loss functions have recently popularized in the self supervised learning literature (Chen et al., 2020b). We combine this in a holistic way with a cross-entropy type loss on the replay data. This allows us to control the representation drift of old classes while maintaining strong classification performance. Note that if a metric learning loss is used alone we needs to perform predictions using a Nearest Class Means Rebuffi et al. (2017) approach, which we show is computationally expensive in the online setting.

Given an input data point $x$, we consider the function $f_\theta(x)$ mapping $x$ to its hidden representation before the final linear projection. We denote the incoming $N$ datapoints by $\mathbf{X}^{in}$ and data replayed from the buffer by $\mathbf{X}^{bf}$. We use the following loss, denoted SupCon (Khosla et al., 2020), on the

incoming data $\mathbf{X}^{in}$.

$$\mathcal{L}_1(\mathbf{X}^{in}) = - \sum_{\mathbf{x}_i \in \mathbf{X}_{in}} \frac{1}{|P(\mathbf{x}_i)|} \sum_{\mathbf{x}_p \in P(\mathbf{x}_i)} \log \frac{\mathrm{sim}\big(f_\theta(\mathbf{x}_p), f_\theta(\mathbf{x}_i)\big)}{\sum_{\mathbf{x}_n \in N \cup P(\mathbf{x}_i)} \mathrm{sim}\big(f_\theta(\mathbf{x}_n), f_\theta(\mathbf{x}_i)\big)} \tag{1}$$

where $\mathrm{sim}(a, b) = \exp(\frac{a^T b}{\tau \|a\|\|b\|})$ computes the exponential cosine similarity between two vectors, with scaling factor $\tau$ (Qi et al., 2018; He et al., 2020). Here we denote the incoming data $\mathbf{x}_i \in \mathbf{X}^{in}$. We use the $P$ and $N$ to denote the set of positive and negatives with respect to $\mathbf{x}_i$ and the positive examples $x_p$ are selected from the examples in $\mathbf{X}^{in} \cup \mathcal{M}$, which are from the same classes as $\mathbf{x}_i$. In the sequel we will consider $\mathbf{x}_n$ selected from $\mathbf{X}^{in} \cup \mathcal{M}$ in two distinct ways: (a) from a mix of current and previous classes and (b) only from

---

**Algorithm 1:** ER-AML

**Input:** Learning rate $\alpha$

**Initialize:** Memory $\mathcal{M}$; Model Params $\theta$ **do**

  **Receive** $\mathbf{X}^{in}$         //Receive from stream
  $\mathbf{X}_{pos}, \mathbf{X}_{neg} \sim \mathrm{FETCHPOSNEG}(\mathbf{X}^{in}, \mathcal{M})$
  $\mathbf{X}^{bf} \sim \mathrm{SAMPLE}(\mathcal{M})$        //Sample buffer
  $\mathcal{L} = \gamma \mathcal{L}_1(\mathbf{X}^{in}, \mathbf{X}_{pos}, \mathbf{X}_{neg}) + \mathcal{L}_2(\mathbf{X}^{bf})$
  $SGD(\nabla\mathcal{L}, \theta, \alpha)$        //Param Update
  $\mathrm{RESERVOIRUPDATE}(\mathcal{M}, \mathbf{X}^{in})$        //Save

**while** *The stream has not ended*

---

classes of the $\mathbf{X}^{in}$. Note that this implicitly learns a distance metric where samples of the same class lie close by. For the rehearsal step, we apply a modified cross-entropy objective as per Qi et al. (2018) which allows us to link the similarity metric from above to the logits.

$$\mathcal{L}_2(\mathbf{X}^{bf}) = - \sum_{x \in \mathbf{X}_{bf}} \log \frac{\mathrm{sim}\big(\mathbf{w}_{c(x)}, f_\theta(\mathbf{x})\big)}{\sum_{c \in C_{all}} \mathrm{sim}\big(\mathbf{w}_c, f_\theta(\mathbf{x})\big)} \tag{2}$$

where $C_{all}$ the set of all classes observed, and $c(x)$ denotes the label of $x$. The above formulation allows us to interpret the rows of the final projection $\{\mathbf{w}_c\}_{c \in C_{all}}$ as class prototypes and inference to be performed without need for nearest neighbor search. We combine the loss functions on the incoming and replay data

$$\mathcal{L}(\mathbf{X}^{in} \cup \mathbf{X}^{bf}) = \gamma \mathcal{L}_1(\mathbf{X}^{in}) + \mathcal{L}_2(\mathbf{X}^{bf}) \tag{3}$$

We refer to this approach as Experience Replay with Asymmetric Metric Learning (ER-AML). We describe the full rehearsal procedure with ER-AML in Algo 1. Note the buffer may contain samples with the same classes as the incoming data stream. The subroutine FetchPosNeg is used to find one positive and negative sample per incoming datapoint in $\mathbf{X}^{in}$, which can reside in either the buffer memory $\mathcal{M}$ or in $\mathbf{X}^{in}$.

## 4.2 NEGATIVE SELECTION AFFECTS REPRESENTATION DRIFT

The selection of negatives for the proposed loss $\mathcal{L}$ can heavily influence the representation of previously learned classes and is analogous to the key issues faced in the regular replay methods where cross entropy loss is applied to both incoming and replay data. A typical approach in this loss for classification may be to select the negatives from any other class (Hoffer & Ailon, 2015). However this becomes problematic in the continual learning setting as the old samples will be too heavily influenced by the poorly embedded new samples that lie close to the old sample representations. To illustrate what is going on

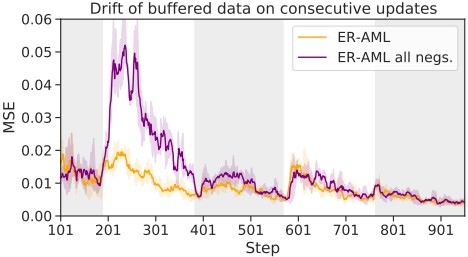

Figure 2: Buffer displacement in a 5 task stream. Background shading denotes different tasks.

in the feature space, consider the case of a ER-AML's $\mathcal{L}_1$ term, which explicitly controls distances between sample representations. $\mathcal{L}_1$ considers the incoming batch samples (containing new classes) as anchors. As the representations from these classes haven't been learned, anchors may end up placed near or in-between points from previous classes (analogous to the illustration in Figure 1). Since the previous classes samples will be clustered together, if we use them as negatives for the incoming sample anchors, the gradients magnitude of the positive term will be out-weighted by the negative terms coming from the new class samples, similar to what is observed in Figure 1. In this case there is a sharp change in gradients norms of the loss w.r.t. the features of previous classes, as we illustrate in Appendix E, which leads to a large change in the representation at the task boundary (and subsequently poor performance). On the other hand if we use only incoming batch examples as

negatives we can avoid this excessive representation drift. We illustrate this in Figure 2 by showing the representations drift at the task boundaries for ER-AML when using negative samples from all classes and when using only classes in the incoming batch. In the context of the model under consideration we measure the one iteration representation drift of a sample $x$ as $\|f_{\theta^t}(x) - f_{\theta^{t+1}}(x)\|$, the output of the network being normalized. We observe that naively applying the proposed loss results in large changes of the learned representation. On the other hand when allowing only negatives from classes in the incoming batch, we see a reduction in this representation drift. In the Appendix 7 we further demonstrate that the accuracy of models trained using ER-AML with only incoming batch negatives can improve the continual learning system performance by a large margin. We emphasize the that ER-AML with all negatives and the regular ER method used for online continual learning suffer from a similar issue and thus lead to similar poor performances, with appropriate negative selection resolving the problem.This is further emphasized in Appendix H where we observe similar poor drift behavior for ER.

### 4.3 CROSS-ENTROPY BASED ALTERNATIVE (ER-ACE)

Having demonstrated the effect of controlling the incoming batch loss in avoiding a drastic representation drift, we now extend it to be applicable to the standard cross-entropy loss typically studied in ER (Aljundi et al., 2019a; Chaudhry et al., 2019). Given an incoming data batch, consider $C_{old}$ the set of previously learned classes and $C_{curr}$ the set of classes observed in the current incoming mini-batch. Denoting $C$ the set of classes included in the cross-entropy loss, we define the $\mathcal{L}_{ce}(\mathbf{X}, C)$ cross-entropy loss as: $\mathcal{L}_{ce}(\mathbf{X}, C) = -\sum_{x \in \mathbf{X}} \log \frac{\mathtt{sim}(w_{c(x)}, f_\theta(x))}{\sum_{c \in C} \mathtt{sim}(w_c, f_\theta(x))}$ where $C \subset C_{all}$ denotes the classes used to compute the denominator. We note that restricting the classes used in the denominator has an analogous effect to restricting the negatives in the contrastive loss. Consider the gradient for a single datapoint $x$, $\frac{\partial \mathcal{L}_{ce}(x, C)}{\partial f_\theta^n} = \mathbf{W}\big((\vec{p} - \vec{y}) \odot \mathbb{1}_{\vec{y} \in C}\big)$. Here $\vec{p}$ denotes the softmax output of the network, $\vec{y}$ a one-hot target, $\mathbb{1}_{\vec{y} \in C}$ a binary vector masking out classes not in $C$, and $\mathbf{W}$ the matrix with all class prototypes $\{\mathbf{w}_c\}_{c \in C_{all}}$. When the loss is applied in the batch setting, it follows that only prototypes whose labels are in $C$ will serve roles analogous to positives and negatives in the contrastive loss. We can then achieve a similar control as the metric learning approach on the learned representations.

Now, our loss applied at each step would be:
$$\mathcal{L}_{ace}(\mathbf{X}^{bf} \cup \mathbf{X}^{in}) = \mathcal{L}_{ce}(\mathbf{X}^{bf}, \, C_{old} \cup C_{curr}) + \mathcal{L}_{ce}(\mathbf{X}^{in}, C_{curr})$$
where $C_{curr}$ denotes the set of the classes represented in the incoming batch and $C_{old}$ denotes previously seen classes that are not presented in the incoming batch, those that we want to preserve their representation. Note this is a straightforward procedure and induces no additional computational overhead. We refer to it as Experience Replay with Asymmetric Cross-Entropy (ER-ACE).

## 5 EXPERIMENTS

We have highlighted the issue of abrupt representation change when new classes are introduced, and propose two methods that address this issue. We now demonstrate that mitigating drift directly leads to better performance on standard online continual learning benchmarks. As in Lopez-Paz et al. (2017); Aljundi et al. (2019a); Chaudhry et al. (2019) we use a reduced Resnet-18 for our experiments, and leave the *batch size* and the *rehearsal batch size* fixed at *10*. This allows us to fairly compare different approaches, as these parameters have a direct impact on the computational cost of processing a given stream.

### 5.1 DATASETS

All benchmarks are evaluated in the single-head setting, i.e. task descriptors are not provided to the model at test time, hence the model performs $N$-way classification where $N$ is the total amount of classes seen.

**Split CIFAR-10** partitions the dataset into 5 disjoint tasks containing two classes each (as in Aljundi et al. (2019a); Shim et al. (2020))

**Split CIFAR-100** comprises 20 tasks, each containing a disjoint set of 5 labels. We follow the split in Chaudhry et al. (2019). All CIFAR experiments process $32 \times 32$ images.

**Split MiniImagenet** splits the MiniImagenet dataset into 20 disjoint tasks of 5 labels each. Images are $84 \times 84$.

## 5.2 BASELINES

We focus our evaluation on replay-based methods, as they have been shown to outperform other approaches in the online continual learning setting Chaudhry et al. (2019); Aljundi et al. (2019a); Ji et al. (2020). We keep buffer management constant across methods : all samples are kept or discarded according to Reservoir Sampling Vitter (1985). We consider the following state-of-the-art baselines:

**ER**: Experience Replay with a buffer of a fixed size. Unlike Aljundi et al. (2019a), we do not leverage the task identifier during training to ensure that rehearsal samples belong to previous classes.

**iCaRL** Rebuffi et al. (2017) A distillation loss alongside binary cross-entropy is used during training. Samples are classified based on closest class prototypes, obtained from recomputing and averaging buffered data representations.

**MIR** Aljundi et al. (2019a) selects for replay samples interfering the most with the incoming data batch.

**DER++** Buzzega et al. (2020) uses a distillation loss on the logits to ensure consistency over time.

**SS-IL** Ahn et al. (2020) learns both the current task loss and the replay loss in isolation of each other. An additional task-specific distillation is used on the rehearsal data.

**GDUMB** Prabhu et al. (2020) performs offline training on the buffer with unlimited computation and unrestricted use of data augmentation at the end of the task sequence.

**iid**: The learner is trained with a single pass on the data, in a single task containing all the classes. We also consider a version of this baseline using a similar compute budget as replay methods (**iid++**)

We note additional baselines such as Lopez-Paz et al. (2017); Chaudhry et al. were shown to perform poorly in this setting by prior work Buzzega et al. (2020) and are thus left out for clarity.

## 5.3 EVALUATION METRICS AND CONSIDERATIONS

Our evaluation includes the metrics and experimental settings used in previous works on online continual learning with a single-head (Aljundi et al., 2019a; Ji et al., 2020; Shim et al., 2020). We provide extra emphasis on anytime evaluation and comparisons of the computation time per incoming batch. We also consider several additional settings in terms of computation and use of image priors.

**Anytime evaluation** A critical component of online learning is the *ability to use the learner at any point* De Lange et al. (2019) . Although most works in the online (one-pass through the data) setting report results throughout the stream Lopez-Paz et al. (2017); Chaudhry et al.; Aljundi et al. (2019b), several prior works have reported the final accuracy as a proxy Aljundi et al. (2019a); Shim et al. (2020). However a lack of anytime evaluation opens the possibility to exploit the metrics by proposing offline learning baselines that are inherently incompatible with anytime evaluation Prabhu et al. (2020).

In order to make sure that learners are indeed online learners, we evaluate them throughout the stream. We define the Anytime Accuracy at time $k$ $(AA_k)$ as the average accuracy on the test sets of all distributions seen up to time $k$. If the learning experience lasts $T$ steps, then $AA_T$ is equivalent to the final accuracy. Finally, we report the *Averaged Anytime Accuracy (AAA)* (Caccia et al., 2020), which measures how well the model performed over the learning experience

$$\text{AAA} = \frac{1}{T}\sum_{t=1}^{T}(AA)_t. \tag{4}$$

**Computation and Memory Constraints** While memory constraints are well documented in previous work, careful monitoring of computation is often overlooked; some methods can indeed hide considerable overhead which can make the comparison across methods unfair. On the other hand this is critical to the use cases of online continual learning. To remedy this, we report for each method the total number of FLOPs used for training. While we cannot fix this quantity as we can for memory (since different methods require different computations), this will shed some light on how different methods compare. Note that we also include in this total any inference overhead required by the models; Nearest Class Mean (NCM) classifiers must compute class prototypes before inference for example. We add this cost every time the model is queried to measure its Anytime Accuracy. Let

$$\text{Mem} = \frac{1}{T}\sum_{t=1}^{T}|\theta_t| + |\mathcal{M}_t|, \quad \text{Comp} = \sum_{t=1}^{T}\mathcal{O}(m(\cdot;\theta_t)), \tag{5}$$

where $\mathcal{O}(m(\cdot;\theta_t))$ denotes the number of FLOPs used at time $t$. Since the same backbone and buffer is used for all methods in this paper, we will focus our constraint analysis on computation

| Method | Data Aug. | $M=5$ AAA | $M=5$ Acc | $M=20$ AAA | $M=20$ Acc | $M=100$ AAA | $M=100$ Acc | Train TFLOPs | Mem. (Mb) |
|---|---|---|---|---|---|---|---|---|---|
| iid | ✗ | - | $62.7_{\pm0.7}$ | - | $62.7_{\pm0.7}$ | - | $62.7_{\pm0.7}$ | 8 | 4 |
| iid++ | ✗ | - | $72.9_{\pm0.7}$ | - | $72.9_{\pm0.7}$ | - | $72.9_{\pm0.7}$ | 16 | 4 |
| DER++ | ✓ | $50.7_{\pm1.1}$ | $31.8_{\pm0.9}$ | $55.6_{\pm1.2}$ | $39.3_{\pm1.0}$ | $60.1_{\pm1.3}$ | $52.3_{\pm1.1}$ | 24 | **(4, 7)** |
| ER | ✗ | $40.0_{\pm0.8}$ | $19.7_{\pm0.3}$ | $45.2_{\pm1.3}$ | $26.7_{\pm1.0}$ | $55.4_{\pm1.4}$ | $38.7_{\pm0.8}$ | **17** | **(4, 7)** |
| ER | ✓ | $45.6_{\pm1.1}$ | $28.4_{\pm1.0}$ | $55.9_{\pm1.2}$ | $40.3_{\pm0.6}$ | $60.3_{\pm1.3}$ | $49.4_{\pm1.3}$ | | |
| iCaRL† | ✗ | $47.0_{\pm0.8}$ | $30.6_{\pm0.8}$ | $55.1_{\pm0.7}$ | $41.7_{\pm0.6}$ | $59.3_{\pm0.6}$ | $45.1_{\pm0.6}$ | (21, 47) | (8, 11) |
| iCaRL† | ✓ | $49.1_{\pm1.0}$ | $33.4_{\pm1.0}$ | $54.4_{\pm0.7}$ | $39.2_{\pm0.8}$ | $56.9_{\pm0.7}$ | $42.3_{\pm0.8}$ | | |
| MIR† | ✗ | $39.3_{\pm1.0}$ | $19.7_{\pm0.5}$ | $44.7_{\pm1.1}$ | $29.7_{\pm0.6}$ | $53.8_{\pm1.7}$ | $43.3_{\pm1.0}$ | 41 | **(4, 7)** |
| MIR† | ✓ | $44.9_{\pm0.9}$ | $29.8_{\pm0.8}$ | $49.7_{\pm1.0}$ | $41.8_{\pm0.6}$ | $54.6_{\pm1.4}$ | $49.3_{\pm0.6}$ | | |
| SS-IL† | ✗ | $42.6_{\pm1.7}$ | $29.6_{\pm0.4}$ | $44.8_{\pm1.8}$ | $35.1_{\pm0.9}$ | $48.1_{\pm2.2}$ | $41.1_{\pm0.4}$ | 19 | (8, 11) |
| SS-IL† | ✓ | $41.1_{\pm1.6}$ | $31.6_{\pm0.5}$ | $47.0_{\pm1.2}$ | $38.3_{\pm0.4}$ | $48.1_{\pm1.7}$ | $47.5_{\pm0.7}$ | | |
| ER-ACE (ours) | ✗ | $\mathbf{53.1}_{\pm1.0}$ | $35.6_{\pm1.0}$ | $\mathbf{58.0}_{\pm0.7}$ | $42.6_{\pm0.7}$ | $\mathbf{61.9}_{\pm0.9}$ | $52.2_{\pm0.7}$ | **17** | **(4, 7)** |
| ER-ACE (ours) | ✓ | $52.6_{\pm0.9}$ | $35.1_{\pm0.8}$ | $56.4_{\pm1.0}$ | $43.4_{\pm1.6}$ | $\mathbf{61.7}_{\pm0.9}$ | $53.7_{\pm1.1}$ | | |
| ER-AML (ours) | ✗ | $49.4_{\pm1.0}$ | $30.9_{\pm0.8}$ | $\mathbf{57.0}_{\pm1.0}$ | $39.2_{\pm1.0}$ | $\mathbf{63.3}_{\pm1.0}$ | $52.2_{\pm1.1}$ | **17** | **(4, 7)** |
| ER-AML (ours) | ✓ | $50.4_{\pm1.3}$ | $\mathbf{36.4}_{\pm1.4}$ | $56.8_{\pm1.0}$ | $\mathbf{47.7}_{\pm0.7}$ | $62.0_{\pm0.9}$ | $\mathbf{55.7}_{\pm1.3}$ | | |
| GDUMB | ✓ | $0_{\pm0.0}$ | $35.0_{\pm0.6}$ | $0_{\pm0.0}$ | $45.8_{\pm0.9}$ | $0_{\pm0.0}$ | $61.3_{\pm1.7}$ | (43, 853) | (11, 14) |

Table 1: split CIFAR-10 results. † indicates the method is leveraging a task identifier at training time. For methods whose compute depend on the buffer size, we report min and max values. We evaluate the models every 10 updates. Results within error margin of the best result are bolded.

| Method | AAA | Acc. | Train TFLOPs | Mem. (Mb.) | AAA | Acc. | Train TFLOPs | Mem. (Mb.) |
|---|---|---|---|---|---|---|---|---|
| iid | - | $19.8_{\pm0.3}$ | 9 | 4 | - | $16.7_{\pm0.5}$ | 59 | 4 |
| iid++ | - | $28.3_{\pm0.3}$ | 17 | 4 | - | $25.0_{\pm0.8}$ | 118 | 4 |
| DER++ | $23.3_{\pm0.5}$ | $15.1_{\pm0.4}$ | 25 | 36 | $21.7_{\pm0.6}$ | $12.9_{\pm0.3}$ | 176 | 217 |
| ER | $24.2_{\pm0.6}$ | $19.8_{\pm0.4}$ | **17** | **35** | $26.2_{\pm0.8}$ | $18.2_{\pm0.5}$ | **118** | **216** |
| iCaRL† | $26.3_{\pm0.3}$ | $17.3_{\pm0.2}$ | 294 | 39 | $24.4_{\pm0.4}$ | $17.1_{\pm0.1}$ | 2097 | 220 |
| MIR† | $23.6_{\pm0.8}$ | $20.6_{\pm0.5}$ | 41 | **35** | $27.2_{\pm0.7}$ | $20.2_{\pm0.8}$ | 294 | **216** |
| SS-IL† | $31.5_{\pm0.5}$ | $25.0_{\pm0.3}$ | 19 | 39 | $\mathbf{29.7}_{\pm0.6}$ | $\mathbf{23.5}_{\pm0.5}$ | 137 | 220 |
| ER-ACE (ours) | $\mathbf{32.7}_{\pm0.5}$ | $\mathbf{25.8}_{\pm0.4}$ | **17** | **35** | $\mathbf{30.2}_{\pm0.6}$ | $22.7_{\pm0.6}$ | **118** | **216** |
| ER-AML (ours) | $30.2_{\pm0.6}$ | $24.3_{\pm0.4}$ | 28 | **35** | $27.0_{\pm0.7}$ | $19.3_{\pm0.6}$ | 200 | **216** |

Table 2: Split CIFAR-100 (left) and Mini-Imagenet (right) results with $M = 100$. For each method, we report the best result between using (or not) data augmentations.

**Data Augmentation** In the settings of Aljundi et al. (2019a); Lopez-Paz et al. (2017); Ji et al. (2020); Shim et al. (2020); Chaudhry et al. (2019) data augmentation is not used. However, this is a standard practice for improving the performance on small datasets and can thus naturally complement most methods utilizing replay buffers. Notably, Prabhu et al. (2020), the offline learning method, utilizes data augmentation when comparing to the above online learners. To avoid unfair comparisons, in our experiments we indicate when a method uses augmentation. When not specified, we treat it as a hyperparameter and report the best performance.

**Hyperparameter selection** For all datasets considered, we withhold 5 % of the training data for validation. For each method, optimal hyperparameters were selected via a grid search performed on a validation set. The selection process was done on a per dataset basis, that is we picked the configuration which maximized the accuracy averaged over different memory settings. We found that for both ER-AML and ER-ACE, the same hyperparameter configuration worked across all settings and datasets. All necessary details to reproduce our experiments can be found in the Appendix.

## 5.4 Standard Online Continual Learning Settings

We evaluate on Split CIFAR-10, Split CIFAR-100 and Split MiniImagenet using the protocol and constraints from Aljundi et al. (2019a); Ji et al. (2020); Shim et al. (2020) . We note in all results each method is run 10 times, and we report the mean and standard error. We first discuss dataset specific results, before analysing the computation cost of each method.

**CIFAR-10** results are found in Table 1 using a variety of buffer sizes. In this setting, we see that both the methods we propose, ER-AML and ER-ACE *consistently outperform* other methods by a significant margin. This result holds in both settings where data augmentation is (or not) used,

outperforming previous state-of-the-art methods MIR and DER++. Shifting our attention to SS-IL, its underperformance w.r.t to ER-ACE highlights the importance of having a rehearsal objective that considers the new classes. In Appendix B, we observe that when applying SS-IL in the online setting: (1) the method performs poorly on the current task, as is it unable to consolidate old and new knowledge, (2) yet mitigates representation drift even on a perfectly balanced stream. The latter is surprising, as the method was designed specifically to address stream imbalance. Finally, we note the offline training baseline G-DUMB cannot satisfy the anytime evaluation criteria.

**Longer Task Sequence** results are shown in Table 2 with CIFAR-100 on the left and MiniImagenet on the right. On both datasets similar findings are observed, our proposed methods match or outperform strong existing baselines. SS-IL performs similarly to our method on mini-imagenet hile having a higher computational and memory cost. As mentioned above, the method struggles to learn the current task, however here the "weight" of the current task is small in the final acc of the 20-task regime. We see that average anytime accuracy is higher for ER-ACE and indeed the anytime curves in Appendix L further illustrate this. Finally, ER-ACE shows relative gains of **35%** in accuracy over ER, without any additional computation cost. For Mini-Imagenet, ER-ACE outperforms the single-pass iid baseline, and nearly reaches the performance of the equal-compute iid baseline.

**Computation Budget** To provide another view of the computational advantages of our proposal we report the accuracy given compute budget over the length of the sequence in Fig 3. When monitoring the computation performed by each baseline, we notice that several methods do not compete on equal footing. First, the use of Nearest Class Mean (NCM) classifiers leads to a significant compute cost, as shown for iCaRL. For our experiments, we evaluate the model after 10 mini-batches (100 total samples), where NCM classifier **must forward the whole buffer** to get class prototypes. We argue that such an approach has disadvantages in the online setting due to poor computational trade-offs. Second, MIR Aljundi et al. (2019a) has an expen-

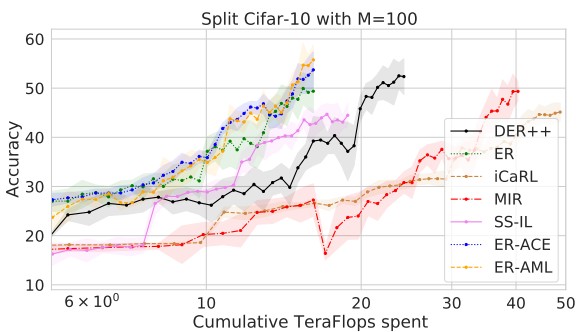

Figure 3: Total Accuracy as a function of TeraFLOPs spent. Here the models are evaluated on all 10 classes, to ensure consistency across timesteps.

sive sample retrieval cost. It remains to show if this step can be approximated more efficiently. Finally, we note that our method, ER-AML has varying compute: for streams with a small number of classes per task (CIFAR10), it can compute the incoming loss leveraging only the incoming data. In other datasets, where an incoming batch may not have at least two samples of each class, an additional cost to forward a buffered point is incurred.

**Evaluation with augmentation** The use of augmentations also permits extra benefits of replay methods particularly in settings where buffer overfitting is more present, e.g. in the small buffer regime. From the results in Table 1, we see that *augmentations provides significant gains* for a large set of methods. It is therefore crucial to compare methods on equal footing, where they can all leverage (or not) data augmentation. For example, *gains reported in Prabhu et al. (2020) over ER completely vanish* when ER is given the same access to augmented data. We note that for Mini-Imagenet, augmentations did not help. We hypothesize that since this is the hardest task the risk of overfitting on the buffer is less severe.

## 5.5 BLURRY TASK BOUNDARIES

Table 3: CIFAR-10 Blurry Task Boundary Experiments

| Method | $M = 20$ | $M = 100$ |
|--------|----------|-----------|
| ER | $32.1_{\pm1.5}$ | $42.7_{\pm2.2}$ |
| DER++ | $31.0_{\pm1.4}$ | $41.7_{\pm1.4}$ |
| ER-AML | $\mathbf{45.6}_{\pm1.2}$ | $\mathbf{55.2}_{\pm1.1}$ |
| ER-ACE | $\mathbf{44.5}_{\pm0.5}$ | $50.2_{\pm1.1}$ |

Next, we explore a setting where the distribution is continuously evolving, rather than clearly delineated by task boundaries (similar to settings considered in Aljundi et al. (2019b)). To do this, we linearly interpolate between tasks over time, resulting in new classes being slowly mixed into the data stream. This experiment is done on Split-CIFAR10, and the interpolation is such that at every timestep, the incoming data batch has on average 2 unique labels (as in the original experiment). We only evaluate task-free methods in this setting: methods like *MIR and SS-IL cannot be used in such setting*. Results in table 3

report the final accuracy, averaged over 5 runs, we report the standard error. We observe our ER-AML and ER-ACE methods perofrm well in this setting. More details provided in Appendix A.2.

## 6 CONCLUSION

We have illustrated how in the online continual learning setting the standard loss applies excessive pressure on old class representations. We proposed two modifications of the loss function, both based on treating the incoming and replay data in an asymmetric fashion. Our proposed method does not require knowledge of the current task and is shown to be suitable for long task sequences achieving strong performance with minimal or no additional cost. We also raise the standard for high quality evaluation in online continual learning by considering a wide number of baselines and metrics.

## 7 REPRODUCIBILITY STATEMENT

We have made several efforts to ensure that the results provided in the paper are fully reproducible. We first provide a detailed codebase from which all the results in this paper are generated. In this codebase, one can find the results of our grid search, as well as optimal hyperparameters for each method and setting. We have provided a Readme file to help guide used to reproduce our results. Details of all hyperparameters are also clearly described in the main paper and particularly in the appendix.

## 8 ACKNOWLEDGEMENTS

Lucas Caccia is funded by Borealis AI. EB and NA are supported by NSERC Discovery Grant RGPIN-2021-04104. We acknowledge resources provided by Compute Canada and Calcul Quebec.

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

## A    EXPERIMENTAL SETUP

In this section we provide additional experiments regarding the baselines and hyperparameters. In all experiments, we leave the **batch size** and the **rehearsal batch size** fixed at **10**, following Aljundi et al. (2019a); Chaudhry et al.. This allows us to fairly compare different approaches, as these parameters have a direct impact on the computational cost of a given run. The model architecture ($\theta$ in Alg. 1) is also kept constant, which is a reduced ResNet-18 used in Lopez-Paz et al. (2017); Chaudhry et al.; Aljundi et al. (2019a;b), where the dimensions of the last linear layer change depending on the input height and width. The model has 1.09M params for the CIFAR experiments and 1.15M params for MiniImagenet. For all datasets considered, we keep the original ordering of the classes, meaning that the first task will always contain the first $k$ classes.

### A.1 Hyperparameters

All results in the paper have been (re)implemented by us, with the expection of GDUMB Prabhu et al. (2020), where results were run from the author's public codebase. For each method a grid search was ran on the possible hparams, which we detail below. We will also described method specific details.

**DER++ Buzzega et al. (2020)** :

- LR : [0.1, 0.01, 0.001]
- $\alpha$ : [0.25, 0.5, 0.75]
- $\beta$ : [0.5, 0.75, 1]

We also tried to implement the DER (not DER++) algorithm described in Buzzega et al. (2020). We found that it did not lead to improvements w.r.t to ER **in the single epoch setting**. Moreover, the setting in the original paper uses a wider Resnet-18. We found that both these differences account for the drop in performance when comparing to the numbers in Buzzega et al. (2020).

Finally, we highlight that in general, methods using distillation (iCaRLRebuffi et al. (2017), SS-IL Ahn et al. (2020), and DER Buzzega et al. (2020)) typically perform better in the onlne setting without it.

**ER Chaudhry et al.** :

- LR : [0.1, 0.01, 0.001]

Note that unlike the ER implementation in Aljundi et al. (2019a), we use a "task-free" implementation. This leads to two differences. First, rehearsal begins as soon as the buffer is not empty. Second, when fetching points in the buffer, **we do not exclude classes from the current task**, as done in MIRAljundi et al. (2019a).

**iCaRL Rebuffi et al. (2017)** :

- LR : [0.1, 0.01, 0.001]

For all "task-based" methods (iCaRL, MIR, SS-IL) we fully leverage the task identified and do not start rehearsal until the second tasks. This typically leads to better performance, especially in the small buffer setting, as it reduces the work of overfitting to the buffer.

**MIR Aljundi et al. (2019a)** :

- LR : [0.1, 0.01, 0.001]

Note that unlike in the original paper, the final results in the paper are on the full training set. In other words, once the hyperparameter cross-validation is done, we train on the validation set. This changes the results slightly from the original paper. Finally, we kept the number of items subsampled from the buffer for the sampling step $(N_c)$ equal to $50$ as in the original codebase.

**SS-ILAhn et al. (2020)** :

- LR : [0.1, 0.01, 0.001]
- `should distill` : [Yes, No]. When turned on, this method also uses the distillation loss as prescribed in Ahn et al. (2020)

As we will see in B, using the

**ER-ACE**

- LR : [0.1, 0.01, 0.001]

To implement the masking loss, we simply use `logits.maskedfill(mask, -1e9)` to filter out classes which should not receive gradient. Using a small constant in this step is equivalent to removing the masked classes from the softmax denominator.

**ER-AML** :

- LR : [0.1, 0.01, 0.001]
- SupCon Temperature : [0.1, 0.2]

## A.2 Blurry Task Boundaries Experiment

Here we provide additional details on the experiment described in Section 5.4. In the original (task based) benchmark, each task comprises 10K samples (or 1K minibatches of 10 samples), so a total of 5K minibatches streamed. For the smooth alternative, at each timestep $t \in \{1, 2, .., 5000\}$ the unnormalized probability of seeing class $c$ is given by

$$p_c(t) \sim \mathcal{N}(\mu_c - t, \frac{N_c}{4})$$

with $N_c$ denotes the number of samples of class $c$, and $\mu_c = \frac{(2c-1)N_c}{2}$. At every timestep we normalize this probability for each class and sample according to a Categorical distribution with these probabilities. The parameters for the mean and variance are chosen so that on average, the model receives 2 unique labels per minibatch of 10 items (as in the original task-based experiment).

In such a setting where there is no notion of the current task, or rather a set of current labels, one cannot use SS-IL, as it needs to leverage a task identifier during training. Through this experiment we show that our method can overcome this limitation, despite sharing some similarities with SS-IL.

## B  An in-depth analysis of SS-IL in the online setting

SS-IL is a related method. In this section, we highlight several key observations when deploying SS-IL in the online setting which are on the other hand not issues for ER-AML and ER-ACE. We then provide several additional experiments, shedding some light on the inner workings of the method.

**SS-IL fails to learn the current task**  As stated earlier, the key difference between SS-IL without distillation and ER-ACE is that in the latter, *the rehearsal loss in unmasked.* In this section, we highlight the problems that occur when using a masked rehearsal loss alongside a masked incoming loss as in SS-IL. We show that since both losses are masked, *the model never learns to classify classses across tasks.* Specifically, there is no objective in which the model learns to distinguish classes in the current task from classes in the previous tasks. As we show in Figure 4, SS-IL is unable to classify samples from the current task in a single-head setting. The method actually performs worse than random chance on samples form the current task. On the other extreme we see that ER does very well on the current task (shifting abrupty the previous representations to accomodate the new task). Finally, we see that ER-ACE strikes a good tradeoff between the two, reaching a reasonable accuracy on the current task without disrupting the learned representations of previous tasks. We note that the same conclusion is reached when using the original SS-IL method with the distillation loss.

**SS-IL does more than correcting for class imbalance**  SS-IL is motivated as a method which addresses the class imbalance issue arising in replay methods. Specifically, when drawing a fixed number of rehearsal points at every epoch, it follows that as more and more tasks are seen, previous classes are underrepresented in the training stream when compared to points from the current tasks.

In this section, we test whether or not the behavior of SS-IL differs from standard Experience Replay *when no class imbalance is present*. In this experiment, we increase the number of rehearsal points sampled at every task such that when combining incoming and rehearsal data, we obtain perfectly balanced training data on average. This is experiment is done on the Split-CIFAR10 benchmark with 2 classes per task, with a minibatch of 10 incoming datapoints. Therefore, we sample $0, 10, 20, 30, 40$ rehearsal points per incoming databatch for the first, second, third, fourth and fifth task.

What we observe is that SS-IL still outperforms regular Experience Replay, suggesting that the method does more than simply addressing class imbalance in the data stream. We report final accuracy in Table 4. SS-IL's performance gap with ER is bigger with small buffer. This is consistent with what we observe for representation drift : methods with larger buffer can better correct for abrupt representation change, making the gap between ER vs ER-ACE and ER-AML smaller. From this we give new insights on the inner workings of SS-IL, namely that it works well because it addresses representation drift rather than class imbalance.

| Method | $M = 20$ | $M = 50$ | $M = 100$ |
|---|---|---|---|
| ER | $21.0 \pm 1.2$ | $25.7 \pm 1.1$ | $37.8 \pm 0.7$ |
| SS-IL | $30.3 \pm 1.0$ | $34.6 \pm 0.8$ | $39.1 \pm 0.6$ |

Table 4: Final Accuracy on split CIFAR-10 with class balanced stream.

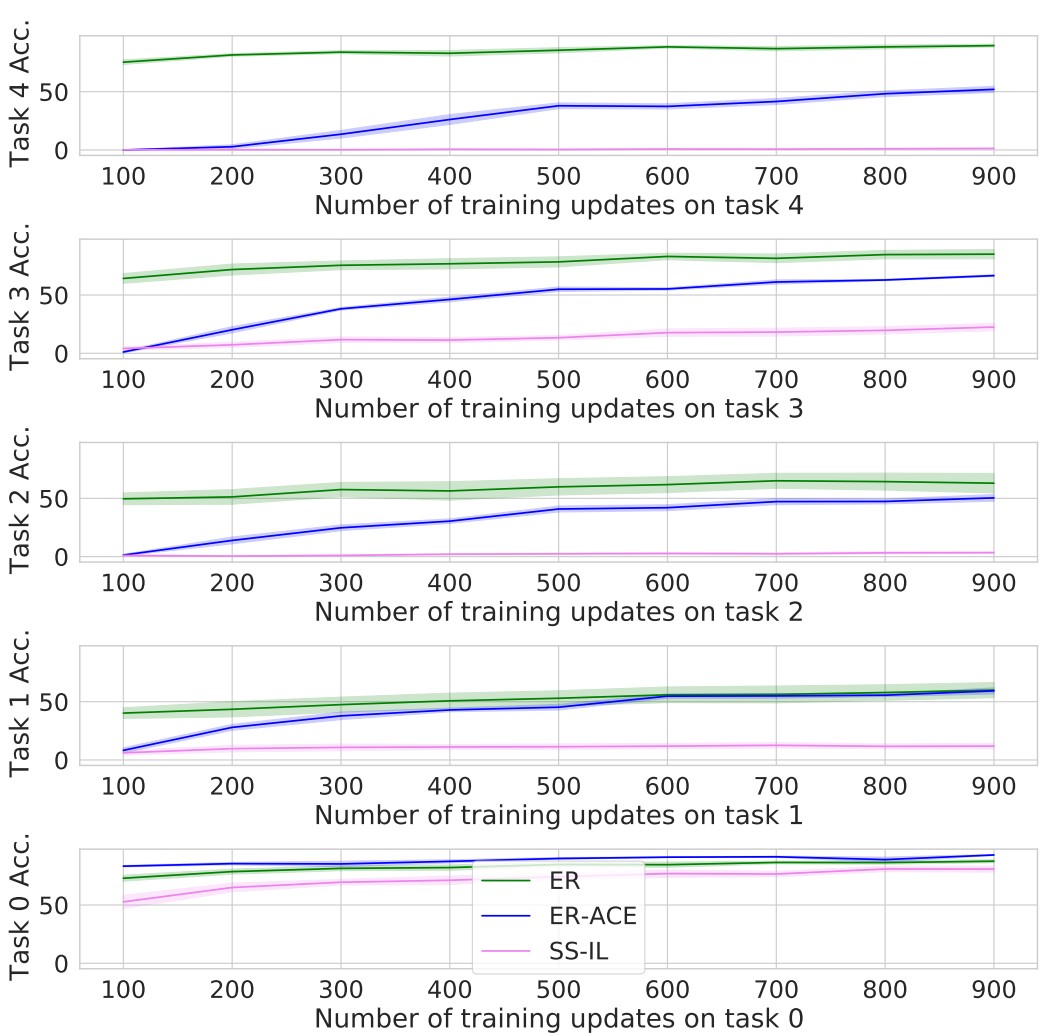

Figure 4: For Split-CIFAR-10, we monitor the performance on the current task observed in the stream for SS-IL, ER, and ER-ACE. ER fits too abruptly current task; ER-ACE incorporates this knowledge slowly; SS-IL barely on the other hand is unable to learn new tasks when they are first observed in the stream

## C OVERFITTING ON BUFFERED SAMPLES

We study the extent to which our proposed method reduces over-fitting to samples stored in the buffer.

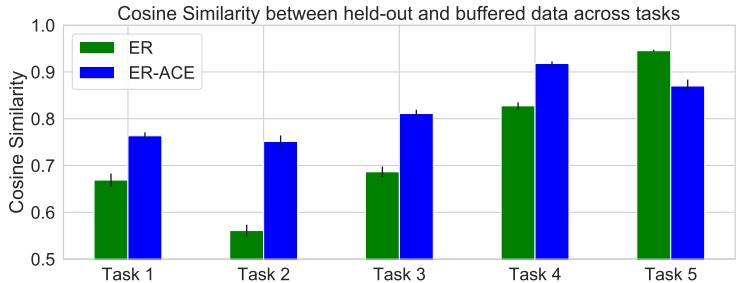

Figure 5: Alignment between buffer and holdout representations. ER-ACE has constantly larger alignment between seen and unseen samples compared to ER especially for older tasks.

A good model fit should yield a learned representation where same class datapoints are aligned, *whether or not they were seen during training.* To evaluate this potential mismatch, we first train a model and compare the representations of a) samples in the buffer $M$ after training and b) held-out samples from the validation set $V$. That is, for each datapoint $x_m \in M$ we find the point $x_v \in V$ with $c(x_m) = c(x_v)$ which maximizes the cosine similarity between $f_\theta(x_m)$ and $f_\theta(x_v)$ . This allows to compare alignment across models, irrespective of their internal scaling. We report the results in Figure 5, where similarity values are averaged over points from the same task. We find that our proposed method, ER-ACE, designed to reduce representation drift also reduces the extent to which the model overfits on the buffer. We observe that for earlier tasks, ER-ACE still retains a strong alignment between rehearsal and held-out data, which is not the case for ER.

## D COMBINING ER-ACE WITH DER++

In this section, we apply our method on top of the strong DER++Buzzega et al. (2020) baseline. For this experiment, we use the same setting as in the DER paper. Specifically, we port our implementation to their public codebase `https://github.com/aimagelab/mammoth`. We keep the default settings for CIFAR-10, using a single pass through the data. We find that combining ER-ACE with DER++ yields additional advantages. Not only do we observe small gains in accuracy, we notice significant gains in forgetting. Results are shown in Figure 6. Forgetting is defined as in Chaudhry et al..

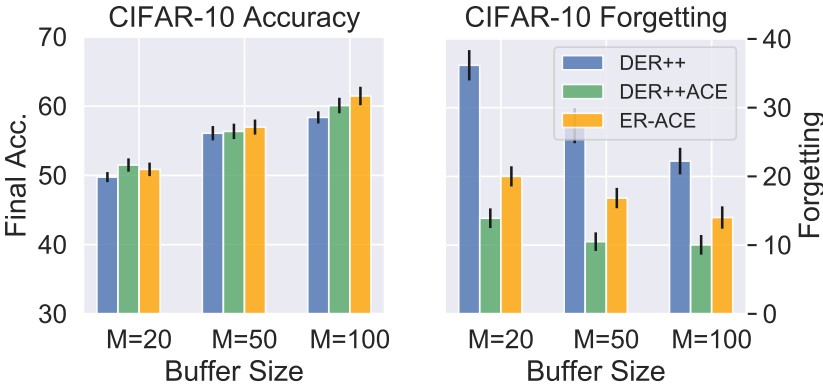

Figure 6: Comparison to Dark Experience Replay (DER). We obtain improved performance and we can enhance the DER method using the ER-ACE approach

| | |
|---|---|
| ER | $(3.2 \pm 1.8) \times 10^{-2}$ |
| ER-AML-Triplet w. All Negs | $(3.0 \pm 0.6) \times 10^{-2}$ |
| ER-AML-Triplet w. Incoming Negs | $(2.5 \pm 0.6) \times 10^{-2}$ |

Table 5: Average Drift (avg distance in feature space) of buffered representations for CIFAR-10 during learning of the second task. We observe similar behavior to ER-AML with SupCon

| | Accuracy ↑ | | | |
|---|---|---|---|---|
| | $M = 5$ | $M = 20$ | $M = 50$ | $M = 100$ |
| iid online | $60.8 \pm 1.0$ | $60.8 \pm 1.0$ | $60.8 \pm 1.0$ | $60.8 \pm 1.0$ |
| iid++ online | $72.0 \pm 0.1$ | $72.0 \pm 0.1$ | $72.0 \pm 0.1$ | $72.0 \pm 0.1$ |
| iid offline | $79.2 \pm 0.4$ | $79.2 \pm 0.4$ | $79.2 \pm 0.4$ | $79.2 \pm 0.4$ |
| fine-tuning | $18.4 \pm 0.3$ | $18.4 \pm 0.3$ | $18.4 \pm 0.3$ | $18.4 \pm 0.3$ |
| ER | $19.0 \pm 0.1$ | $26.7 \pm 0.3$ | $36.1 \pm 0.6$ | $41.5 \pm 0.6$ |
| ER-AML Triplet | $33.0 \pm 0.3$ | $40.1 \pm 0.4$ | $46.0 \pm 0.5$ | $49.8 \pm 0.5$ |
| ER-AML SupCon | $33.0 \pm 0.2$ | $\mathbf{41.9 \pm 0.1}$ | $\mathbf{48.3 \pm 0.2}$ | $\mathbf{51.9 \pm 0.3}$ |

Table 6: Ablation comparing ER-AML with triplet loss to ER-AML with SupCon. We observe both improve over ER but SupCon has better performance in larger buffer sizes

## E   GRADIENT NORM

Figure 7 shows the gradients norms of the features of previous classes in a stream of two tasks. Note how for normal ER, at the task switch the gradients of the previous classes features are suddenly very high leading potentially to large drift on these features.

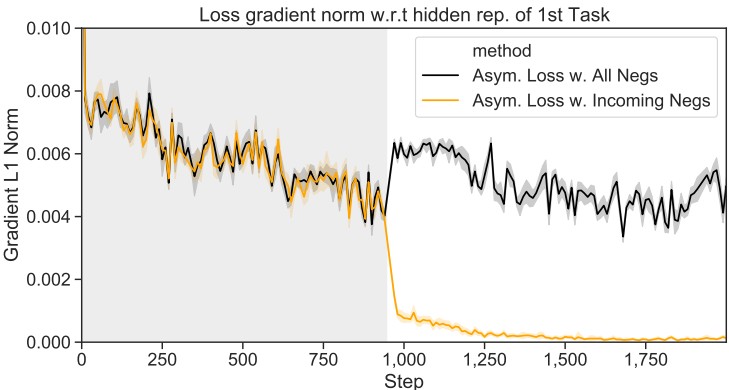

Figure 7: Gradient's norm for first task features in a two task learning scenario. We observe a sharp increase when all negatives are used and decrease using only incoming negatives.

## F   ER-AML WITH TRIPLET LOSS

We observe similar behavior for ER-AML implemented with the triplet loss in terms of the importance of negative selection on drift as illustrated in Table 5. We also ablate ER-AML based on SupCon and Triplet in Table 6 finding the former outperforms in settings with higher buffer sizes, but that both outperform ER.

## G   ABLATIONS NEGATIVE SELECTION

As discussed in the main paper, the selection of negatives is a critical aspect of ER-AML and motivates ER-ACE. To further illustrate this we ablate the performance of ER-AML when all possible negatives are used versus the prescribed negative selection strategy (using only classes in the incoming batch). The results are shown in Table 7. We observe that performance of ER-AML with all negatives is similar to but slightly better than ER, while use of well-selected negatives greatly improves performance.

| | Accuracy (↑ is better) | | Forgetting (↓ is better) | |
| --- | --- | --- | --- | --- |
| | $M = 20$ | $M = 50$ | $M = 20$ | $M = 50$ |
| ER | $26.7 \pm 0.3$ | $36.1 \pm 0.6$ | $47.1 \pm 0.8$ | $37.6 \pm 0.9$ |
| ER-AML(all negatives) | $28.5 \pm 0.3$ | $41.4 \pm 0.4$ | $56.7 \pm 0.6$ | $35.0 \pm 0.4$ |
| ER-AML(incoming negatives) | $\mathbf{41.9 \pm 0.1}$ | $\mathbf{48.3 \pm 0.2}$ | $\mathbf{33.6 \pm 0.2}$ | $\mathbf{25.8 \pm 0.3}$ |

Table 7: Ablation of ER-AML with all negative selection versus negatives selected from incoming classes. We use the CIFAR-10 dataset. We observe that performance of ER-AML with all negatives is similar to but slightly better than ER, while use of well-selected negatives greatly improves performance.

## H  ADDITIONAL DRIFT RESULTS

We showed in Figure 2 that the selection of negatives has a significant impact on the amount of representation change. Here we show that a similar behavior is observed with ER vs ER-ACE.

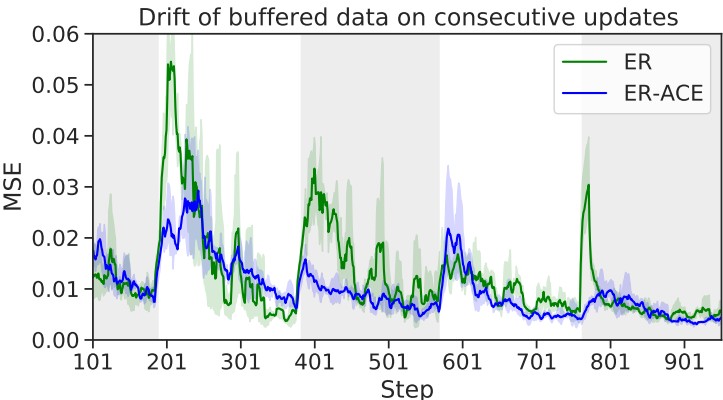

## I  ANALYSIS OF THE REPRESENTATIONS DURING THE SECOND TASK

In this section we take a closer look at the model's internal representation during the learning of the second task for different methods. This experiment replicates the setup illustrated in Figure 1 (split-CIFAR-10 with $M = 20$). For each method, the figures for all iterations were projected together to ensure that the figures are comparable across timesteps. All methods were initialized starting from the same base model trained on the first task. The dotted representations shown for each class come from held-out samples.

We start by looking at the representations obtained at the begining of the second task. We see that for all three methods, (i) the prototypes of the classes from the first task (Class 0 and Class 1) are well placed, while the other prototypes are placed at random since they are not trained.

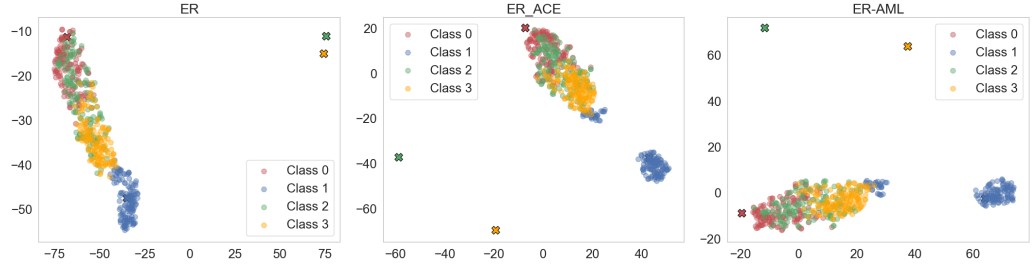

Figure 8: 1 Training Iteration on the Second Task

After 100 training iterations, we see that for ER, the prototypes of the old classes have been signifi-cantly displaced and are far from the points of similar class. This is not the case for the latter two

methods; for ER-ACE and ER-AML, the model is beginning to separate de classes from one another, and the class prototypes are near their respective classes.

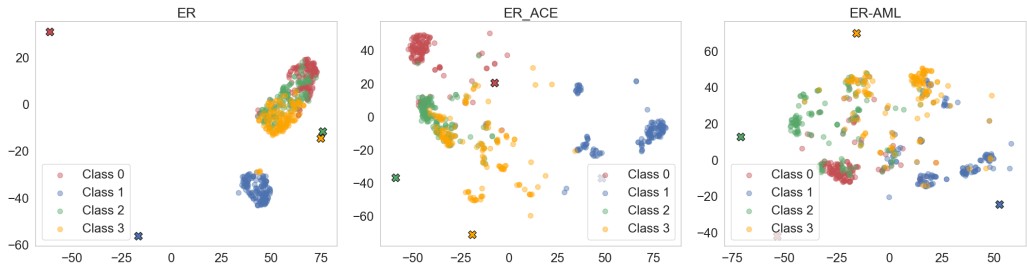

Figure 9: 100 Training Iterations on the Second Task

After 400 training iterations, ER still struggles to align the class prototypes with the respective classes. ER-ACE has already well clustered the respective classes. ER-AML, continues to cluster the classes together, however does not do it as fast as ER-ACE.

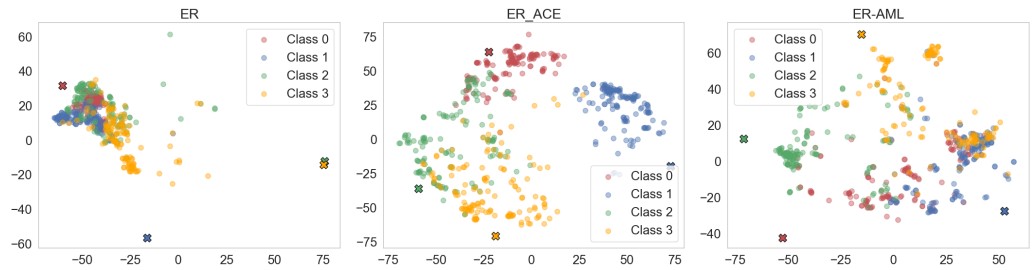

Figure 10: 400 Training Iterations on the Second Task

At the end of the second task, ER-ACE and ER-AML have successfully clustered the classes **and** aligned their respective prototypes with the clusters. As for ER, while the data is clustered, the prototypes are not properly aligned with class clusters. Moreover, we still see a strong overlap between prototypes of Class 2 and 3.

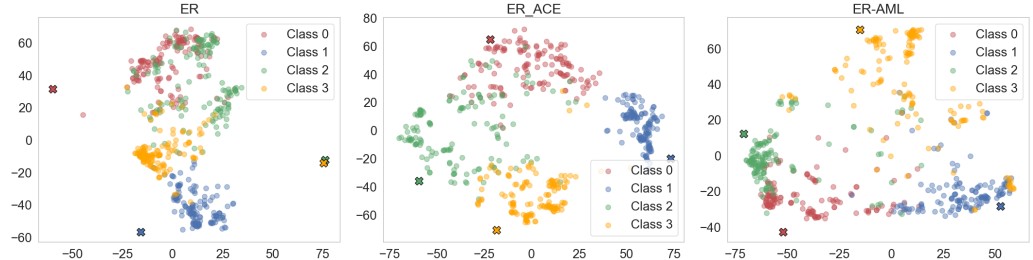

Figure 11: End of the Second Task

## J  ADDITIONAL BLURRY TASK BOUNDARIES EXPERIMENTS

Here we provide blurry task results for varying levels of task overlap. To give an idea of how much the tasks overlap, we report the average number of unique classes per incoming minibatch (MB): a small number means that the tasks are well separated. A high number means that there is a strong overlap. In the fully i.i.d setting, this number would be maximized. On the other hand, when this equals 1, each data class is streamed one after the other.

Experiments are performed again on CIFAR-10 with $M = 20$. We use augmentations to fairly compare with DER++. Results are averaged over 5 runs.

| Method | Avg. unique classes per MB | | | | |
|---|---|---|---|---|---|
| | 1 | 2 | 3 | 4 | 5 |
| ER | 23.1 | 25.7 | 26.3 | 31.1 | 34.4 |
| DER++ | 20.3 | 31.1 | 31.4 | 37.3 | 34.4 |
| ER-ACE | 32.8 | 36.2 | 36.8 | 41.7 | 44.5 |
| ER-AML | **34.0** | **40.4** | **46.0** | **47.6** | **47.9** |

We see that through a wide range of different blurriness levels, our methods show strong improvement over other task-free baselines

## K EXPERIMENTS WITH LIMITED TRAINING DATA AVAILABLE

Next, we evaluate the methods above using varying percentages of the training data from the second task onwards (we use all the data for the first task so the model has converged to a reasonable solution before the first distribution shift). Moreover, we augment the rehearsal batch size for ER, ER-ACE and ER-AML to 20, so that their compute cost equals DER++. This is again on CIFAR-10, $M = 20$. Results averaged over 5 runs.

| Method | % of Data Used | | | |
|---|---|---|---|---|
| | 5% | 10% | 25% | 50% |
| ER | 17.3 | 22.5 | 28.0 | 33.2 |
| DER++ | 17.4 | 19.9 | 24.8 | 32.8 |
| SS-IL | 15.2 | 21.7 | 28.6 | 31.9 |
| ER-ACE | **20.5** | **25.4** | **31.2** | 36.1 |
| ER-AML | 18.1 | 24.3 | 31.0 | **38.7** |

Again, we see that the proposed methods outperforms the baselines suggested above.

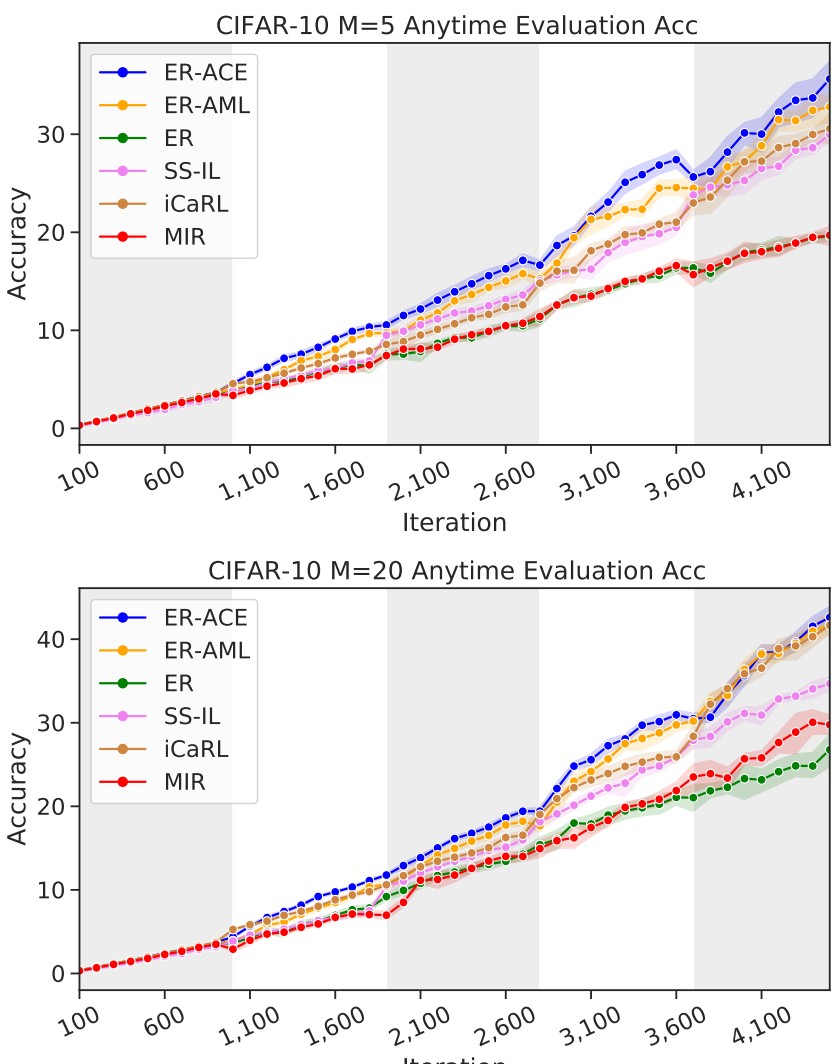

## L    ADDITIONAL RESULTS

In this section we provide full results (shown in the figures below) for various memory sizes on all three datasets considered, i.e. Split CIFAR-10, Split CIFAR-100 and Split MiniImagenet, with and without data augmentation. The results largely align with those presented but also illustrate the anytime performance.

### L.1    ANYTIME EVALUATION WITHOUT DATA AUGMENTATION

### L.2    ANYTIME EVALUATION WITH DATA AUGMENTATION

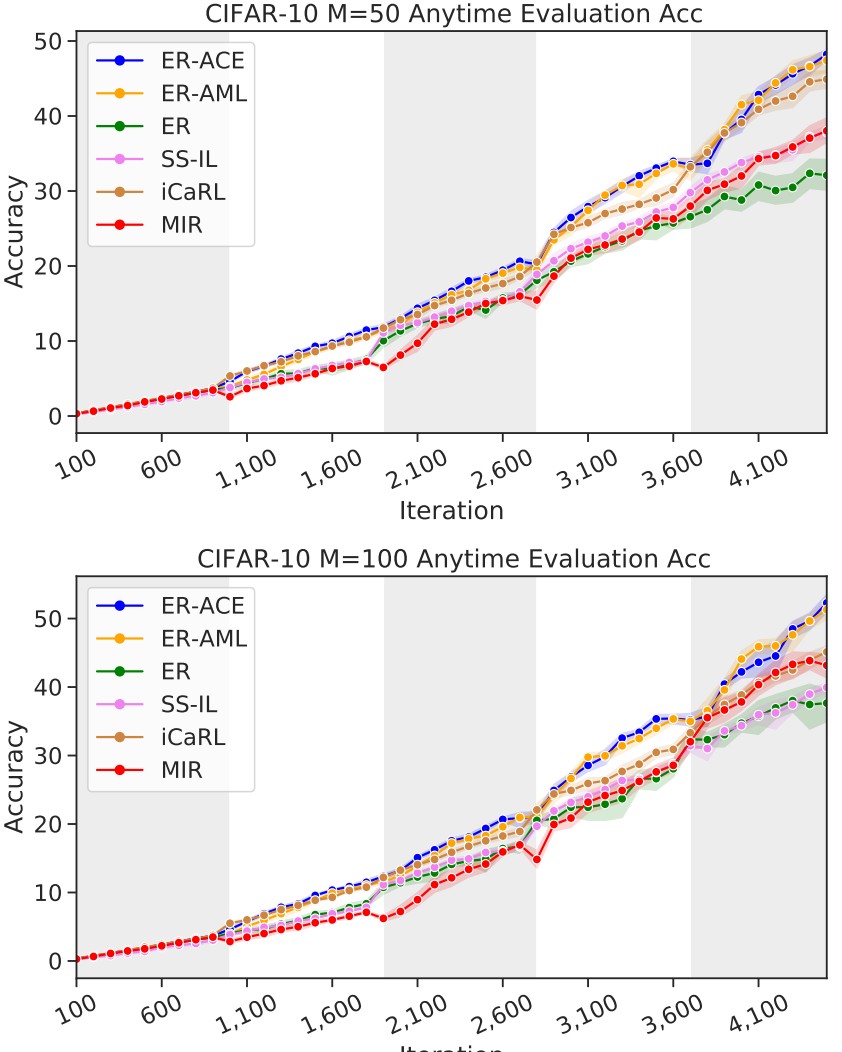

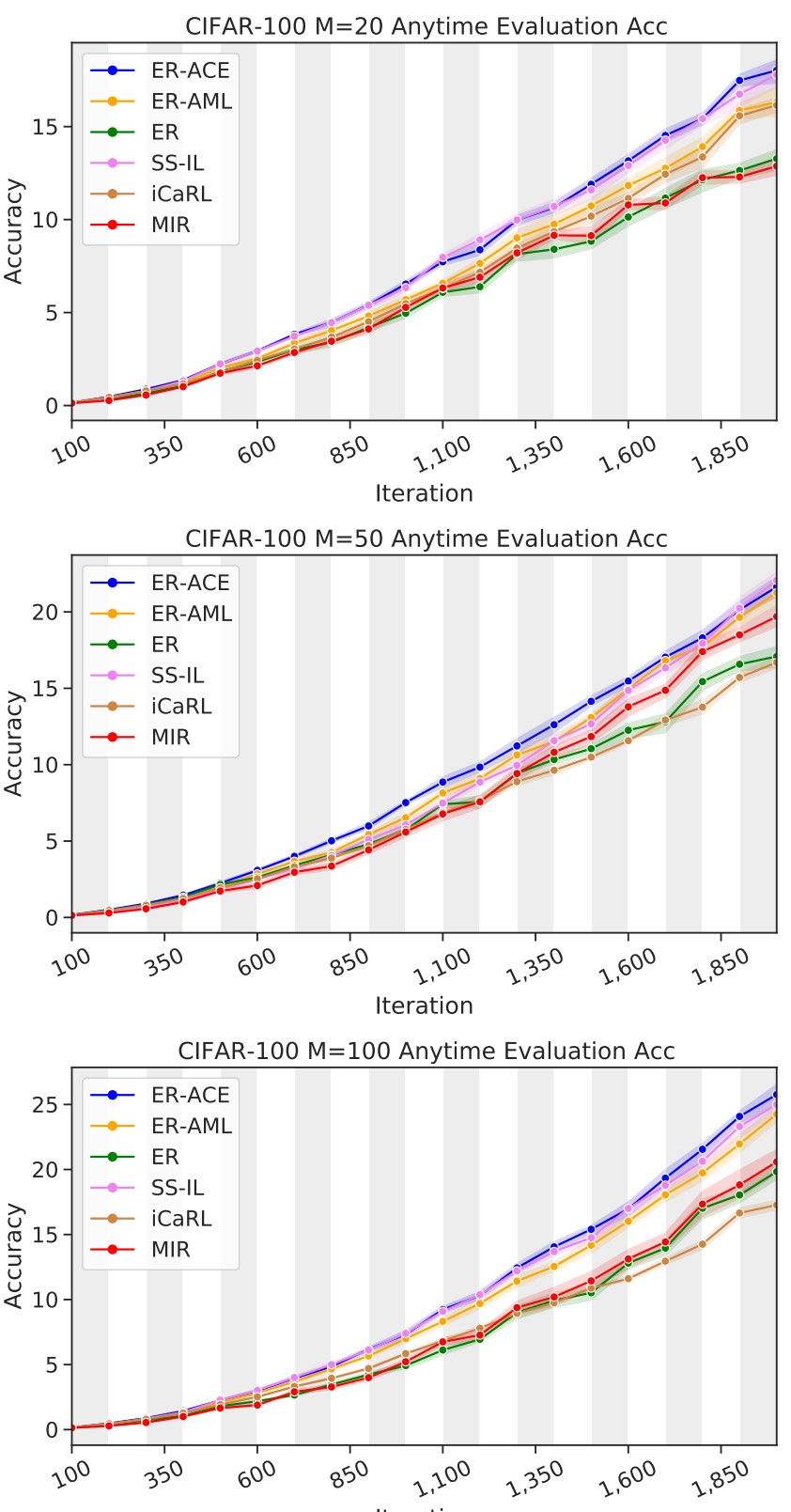

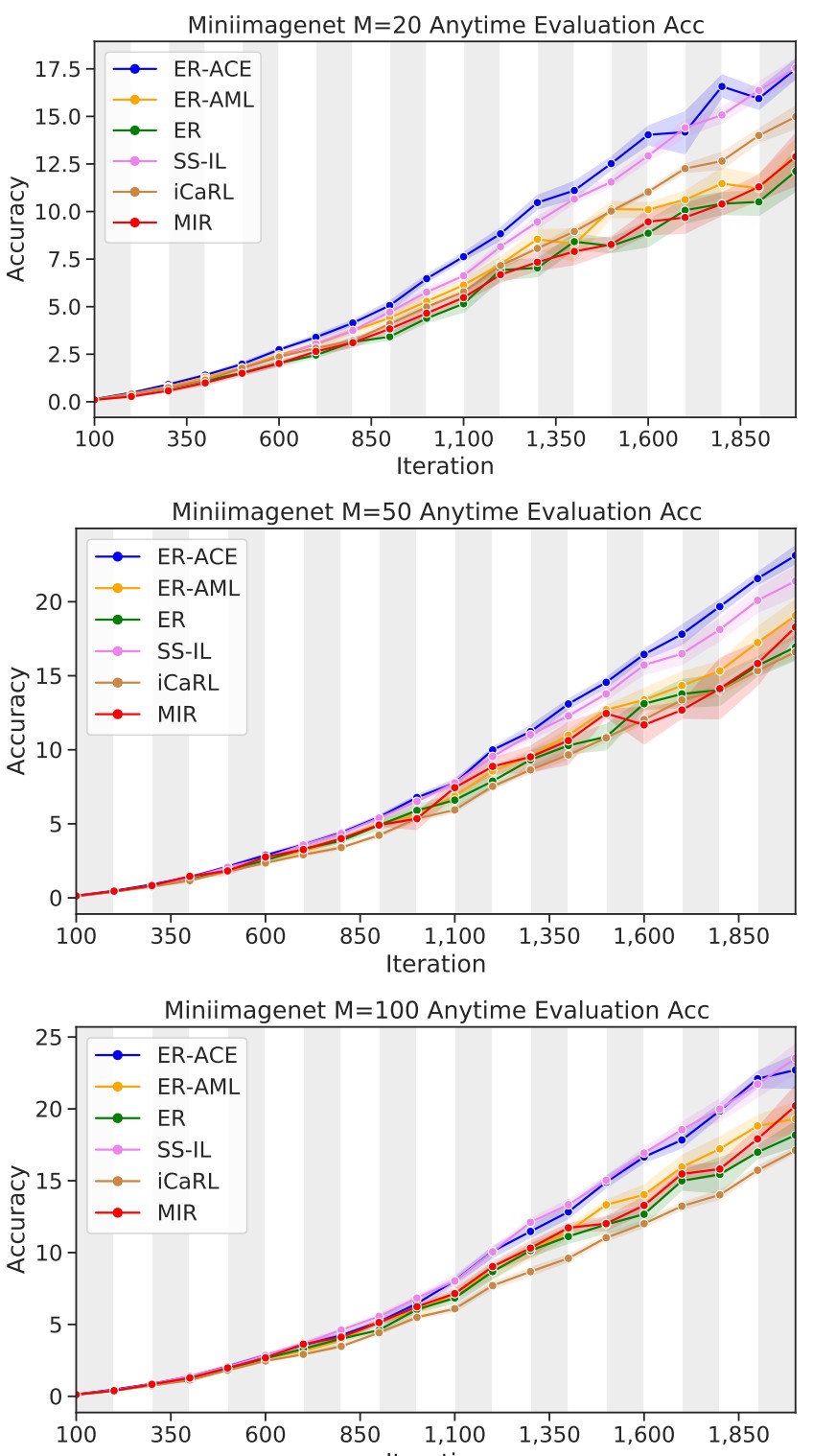

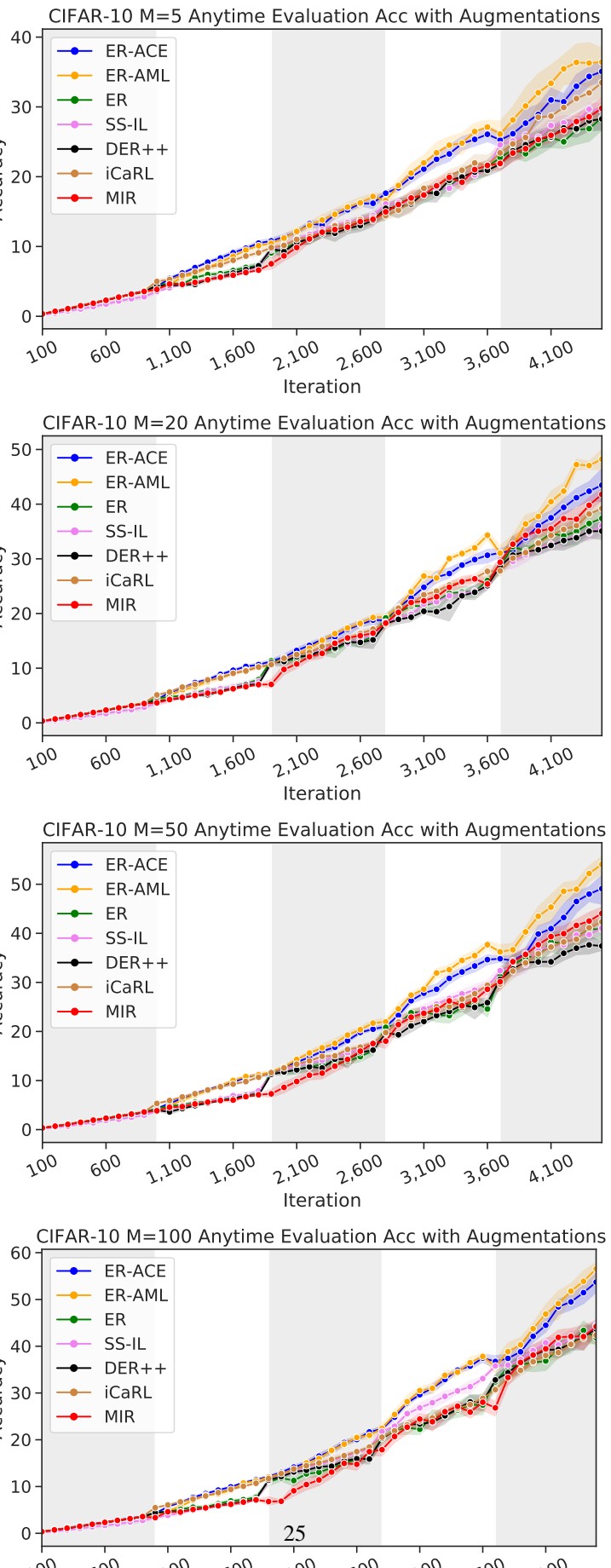

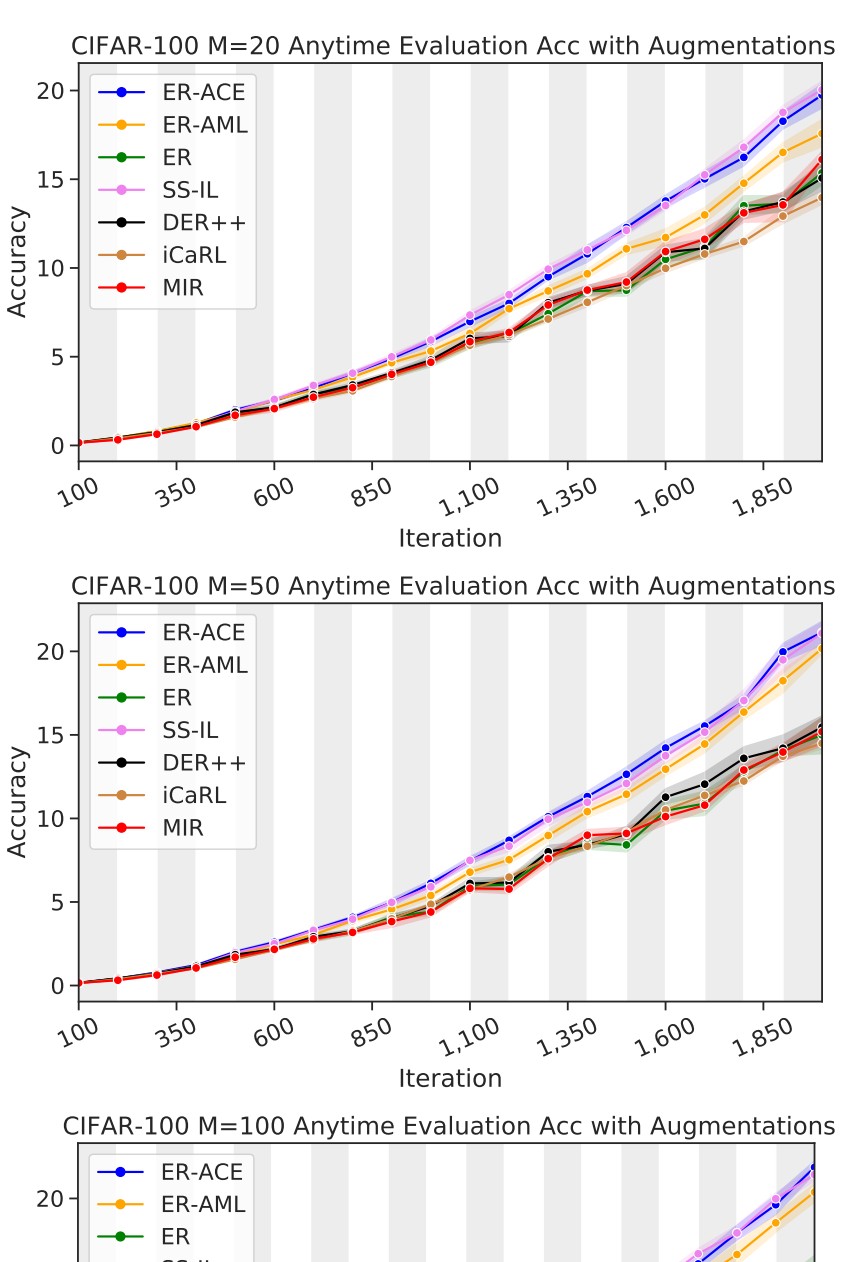

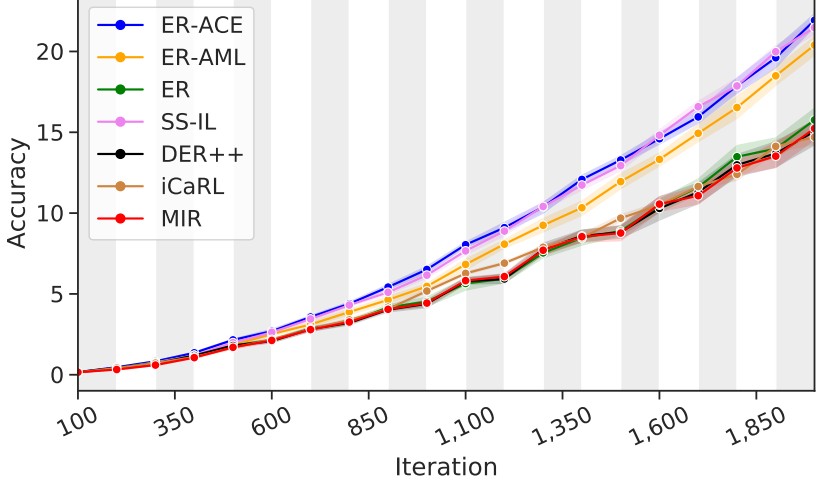

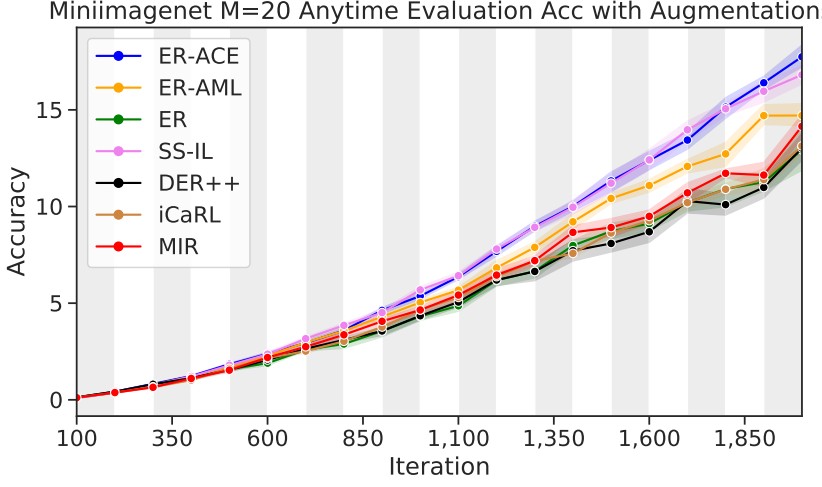

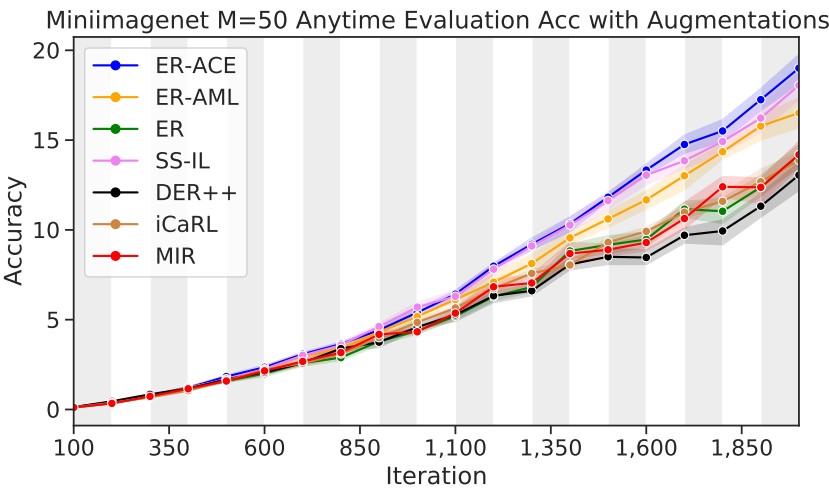

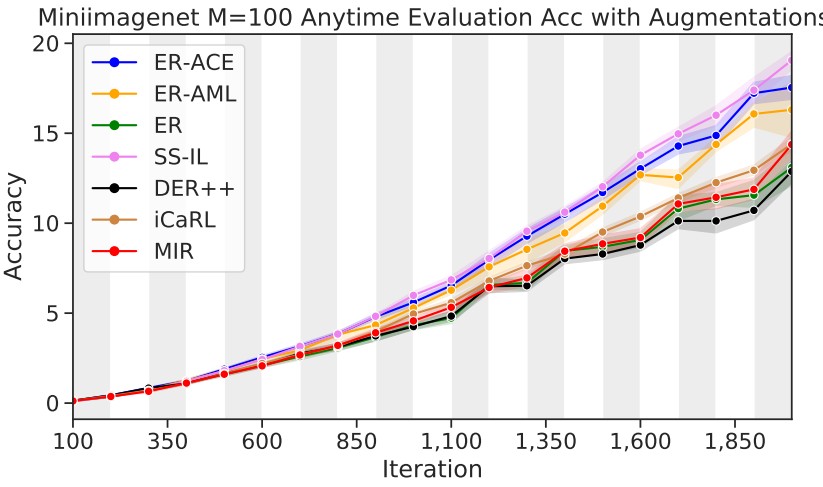

