# OpenReview forum: "New Insights on Reducing Abrupt Representation Change in Online Continual Learning"
_ICLR.cc/2022/Conference — ICLR 2022 Poster_

### Official Review · Reviewer_mCT3 · 2021-10-29

**Correctness:** 3
**Technical Novelty And Significance:** 2
**Empirical Novelty And Significance:** 2
**Recommendation:** 3
**Confidence:** 5

**Main Review:**

Pros:
+ The explanation of catastrophic forgetting from the perspective of abrupt representation drift is somewhat novel. The observation that the knowledge of old classes is recovered by stored samples after disrupted is interesting.
+ Some new metrics are proposed, such as Averaged Anytime Accuracy, Total FLOPs, which make the comparison more rigorous.

Cons:
+ Some important related methods of online continual learning are not included in comparison, e.g. [1, 2, 3]. In particular, [3] also uses a SupCon loss just as ER-AML.
+ The paper claims old tasks suffer from a significant drop in performance when learning new tasks (Fig 1 right). However, this statement is not quite convincing, since the acc drop is not entirely related to representation drift. Some previous work [4, 5] confirm that forgetting is more likely to occur at the layers near the output, especially the last fc layer. Moreover, [6] also finds that adopting metric learning on representations results in less forgetting. To make it clear that forgetting happens at the last fc layer or the whole network, I suggest using NCM classifier to calculate acc here, or another better metric.
+ The motivation of using metric loss is not clear. Compared to [3, 6], which also use metric loss for CL, the paper doesn’t provide enough special motivations or insights to use metric loss for reducing abrupt representation drift. What’s more, the idea to calculate cross entropy loss with replay batch only has already been used in [7]. The proposed losses need more theoretical explanation for why/how they work.
+ Compared to SS-IL, the improvement is not significant.
[1] Online Class-Incremental Continual Learning with Adversarial Shapley Value. AAAI 2021
[2] Graph-Based Continual Learning. ICLR 2021
[3] Supervised Contrastive Replay: Revisiting the Nearest Class Mean Classifier in Online Class-Incremental Continual Learning. CVPR Workshops 2021:
[4] Large Scale Incremental Learning. CVPR 2019
[5] Learning a Unified Classifier Incrementally via Rebalancing. CVPR 2019
[6] Semantic Drift Compensation for Class-Incremental Learning. CVPR 2020
[7] Discriminative Representation Loss (DRL): Connecting Deep Metric Learning to Continual Learning. arxiv.org/abs/2006.11234


**Summary Of The Paper:**

In the online continual learning setting, representation of previous classes will change over time, especially at the task boundary. The authors present a new explanation for the abrupt representation change when applying experience replay (ER) algorithm. They claim that without sufficient training, the representation of new samples is initially mixed with that of previous classes, which causes a sharp increase of gradient resulting shift. To reduce the negative impact of negative new samples, they modify the ER by leveraging asymmetric losses on current data and replay data separately.

**Summary Of The Review:**

Overall, this paper proposes an interesting phenomenon in online continual learning. However, the relation between the solution and the insights are not strong. Besides, the empirical improvements are marginal.

---

> ### Author Response · Authors · 2021-11-12
> **Summary Response to Reviewer mCT3**
>
> Thank you for the careful read of our paper and the insightful suggestions as to the related works! We will first address briefly each of the 4 points raised and go into a more in-depth discussion in separate threads.
>
> 1. Thank you for the suggestions in the related works, specifically for  [3] which is concurrent work, is relevant (yet distinct) to our setting. We note that [2] and [3] are a good example as to why a careful monitoring of compute is important, and a significant contribution of our paper: compared to ER and ER-ACE, papers [2], and [3] use **49x** and **12x** more MFLOPS, a significant increase! Our methods strike a better accuracy / compute tradeoff, critical for the online setting. More comparisons on these papers below, we will also update the related work.
>
>
> 2. Your observation is correct; representation drift will lead to forgetting in the last layer. However, we show in the paper that for Online CL, the root cause of forgetting comes from large gradients at task boundaries (Fig. 1) which, to the best of our knowledge, is a novel finding.
> Per your suggestion, we reproduce Fig. 1 with NCM, and the **drop in accuracy for ER vs ER-ACE is still severe**. More discussion below.
>
> 3. Thank you for pointing out the relevant papers! Note that we do not claim that the use of a metric loss in CL is novel : **the novelty lies in the careful selection of negative samples facilitated by it, which is responsible for the performance gains** (Appendix G). It is for this reason that we use metric losses, as it gives us an explicit control over what samples are compared to each other, which in turns gives us better control over representation drift. Lastly, we do not compute the replay loss in isolation; doing so is detrimental (Appendix B)
>
>
> 4. We respectfully disagree; we show significant gains over SS-IL in 2 of 3 datasets considered, and use less compute and memory. Moreover, our method works in more realistic task-free settings, which SS-IL cannot handle.

---

> > ### Author Response · Authors · 2021-11-12
> > **Detailed Comparison with additional relevant works**
> >
> > We thank you again for pointing out some relevant papers! While these baselines are not in the paper, please let us provide you with some information enabling a fair comparison with our proposed methods. We will restrict our analysis to CIFAR-10, as it is the only setup common to all papers. Moreover, we pick `M=100` as it better illustrates scaling issues with other methods.
> >
> > 1. Aser [1] : Using the public codebase provided by the authors, we were able to estimate the training cost of this approach. The method proposes a novel method to retrieve and fill the buffer, and with this method comes additional forward passes through the model, giving a compute cost of 2.3x that of ER-ACE, and our method **still outperforms Aser by 8% (52.2 vs 43.5)**
> >
> >
> > 2. Graph Based CL [2]: This method performs on-par with ER-ACE. However, this high performance comes at a very expensive computation cost : since the full buffer needs to be forwarded through the model at each timestep, the total MFLOPs is **49x the one of ER-ACE**! It follows that for the same compute our method performs favorably.
> >
> >
> > 3. SCR [3]: This (concurrent) method is the most relevant to our work : it obtains state-of-the-art accuracy for CIFAR-10 `M=100` with 65.7% accuracy. However, once again, the compute cost of this method is high compared to the baselines : since it forwards 2 augmented views per image, uses a large rehearsal batch size, and a NCM classifier, the overall training cost is over **12x** that of ER-ACE. Reducing SCR’s rehearsal batch size such that compute is only **6x** ER-AML’s, the performance drops to **52.6 vs ER-AML’s 55.7**. In all cases, our methods strike a significantly better accuracy vs compute tradeoff. We also emphasize the use of SupCon (which we do not claim to be the novelty of our work) is different here as it is applied to the buffer as well and is not used for the purpose of controlling the representation drift, as we do.

---

> > ### Author Response · Authors · 2021-11-12
> > **Representation Drift vs Last FC Layer**
> >
> > Thank you for connecting our work to that of the offline class-incremental learning community! There are indeed some interesting parallels to be drawn. Indeed, other work has shown that forgetting in various senses is more severe as one goes deeper in the network and towards the top, however this effect is orthogonal from the one we emphasize here: in the *online* class-incremental setting when the loss function has inappropriate contrasts, this leads to large gradients resulting in large representation drifts at task boundaries. In comparison, most of the offline class-incremental paper you linked (as well as SS-IL) argue that forgetting in the last FC layer is due to a class imbalance in the stream. We found that in the online setting, this problem is overshadowed by the issue of abrupt representation change. In fact, in Appendix B Table 4 we show that even on balanced streams ER does not perform better.
> >
> > Per your suggestion, we reproduced the experiment in  Figure 1 with a NCM classifier. We see that just a single update on the second task causes the task 1 accuracy to drop from **92% to 64%** for ER vs **92% to 86.1%** for ER-ACE (5 run avg). Therefore, drift does not occur solely in the last layer, the standard CE classifier causes the representation to be problematic in the online class incremental setting.
> >
> > Lastly, we note that ER-ACE has a direct effect on how the internal representations are clustered during learning. This is demonstrated in Appendix I.

---

> > ### Author Response · Authors · 2021-11-12
> > **Motivation for the Metric Loss**
> >
> > We appreciate the line of work in CL which also uses a metric loss. You mention that [6] shows that metric losses lead to lower forgetting. **We do not observe this in our online setting**. In appendix G, we see that our metric loss used naively can actually lead to an increase in forgetting! It is only when careful negative selection is employed that forgetting is low.
> >
> > Regarding the motivation of our approach : as shown in Figure 1, we want to control how representations from different tasks interact with each other. Metric losses are a great way to do this, since they provide an explicit control over which samples are compared together. This is why we use such losses : to carefully control what samples are compared with the ones from previous tasks. Again, controlling the negative samples plays a crucial role in the effectiveness of our approach.
> >
> > Regarding motivation specific to drift : Fig. 2 in the paper clearly illustrates that our approach to negative selection has a direct impact on representation drift.
> >
> > On computing the replay loss on the replay data only : ER-ACE uses samples across all tasks to compute the rehearsal loss. This is in fact the biggest difference with SS-IL. In Appendix B, we show that when isolating the rehearsal loss (as well as the incoming loss) the model never learns to classify samples across tasks : its accuracy on the current task is actually **worse than random chance**. It is therefore crucial to unmask the rehearsal loss.

---

> > ### Author Response · Authors · 2021-11-12
> > **Gains over SS-IL**
> >
> > Our contribution over SS-IL are two-fold. First, as stated previously, SS-IL (as well as the other offline class-incremental papers you mention) identify as the root cause of the heightened forgetting in the last FC layer to be data imbalance (which results from the current task being overrepresented). We actually found this to be untrue in the online setting : we tried training ER on a balanced stream, and it does not lead to improvements (Appendix B Table 4).
> >
> > Moreover, in this same setting **SS-IL still outperforms ER, even if the root problem it attempts to solve is not present anymore**.  This suggests that SS-IL addresses other issues as well, namely the representation change discussed in this paper. Therefore, we believe our paper offers new insight specific to the online setting (therefore not addressed in other work)
> >
> > Furthermore, removing components from SS-IL which do not affect representation drift (distillation loss and masked rehearsal loss) does not deter its performance, which again points towards representation change being a root cause of forgetting in online CL.
> >
> > Finally, our ER-ACE, while being (a) simpler, (b) fully task-free and  (c) more computationally and memory efficient, always outperforms or performs similarly to SS-IL, which we argue is significant.

---

> > > ### Comment · Reviewer_mCT3 · 2021-12-01
> > > **Thanks for your feedback**
> > >
> > > Your finding is interesting. However, since your paper offers a new insight specific to online setting, why the proposed methods can not remarkably outperform SS-IL. In fact, the proposed methods just outperfrom SS-IL on split CIFAR-10, which is small set with simple settings. For bigger datasets with complex setttings (Split CIFAR-100 and Split MiniImagenet),  it is just match SS-IL. So, It seems that the finding and proposed methods do not bring significant progress on Online Continual Learning.

---

> > > > ### Author Response · Authors · 2021-12-02
> > > > **Thanks you for engaging**
> > > >
> > > > Thank you for your response and engagement.
> > > >
> > > > First, there is **significant value in explaining and understanding why something works,** which is **exactly** what we do : we show that disruptive gradients at task boundaries are responsible for catastrophic forgetting in this setting, a **novel finding** not present in the SS-IL paper, *which does not even consider the online CL setting.*
> > > >
> > > > Based on this insight we developed ER-ACE. Again, besides **performing on par or better** than SS-IL, our method has strong advantages:
> > > >
> > > > 1. It is more computationally and memory efficient. SS-IL uses a distillation loss, which is orthogonal to our motivation and even that of SS-IL and requires extra forward passes, storing an old model, and additional hparams to tune. By removing this additional loss we can compare both methods on equal compute & memory budget. SS-IL's final accuracy drops to **21.3**±0.2 **& 21.8**±0.3 for CIFAR-100 and Mini-Im, compared to **25.8**±0.4 **& 22.7**±0.6 for ER-ACE. Therefore ER-ACE is better on CIFAR-100 and Mini-Im.
> > > >
> > > > 2. SS-IL requires a task-id. In many realistic scenarios, new classes are introduced over time without a clear task delineation. SS-IL cannot be used in this setting, while our methods can. Please see section 5.5 of the main paper.
> > > >
> > > > 3. SS-IL fails to learn the current task in the online setting. We provide an extensive analysis of this in Appendix B, showing that SS-IL's accuracy on the current task is sometimes worse than random chance, a degeneracy from which ER-ACE does not suffer.
> > > >
> > > > Additionally, we disagree in the dismissal of the difficulty of CIFAR-10 for the online CL setting, it is used as a key benchmark dataset in all the papers you have mentioned [1,2,3] and notably a lot of methods such as GEM fail in this sequence [†].  We show considerable gains in this setting over SS-IL because, as pointed out in Appendix B, **SS-IL fails to learn the current task, which is also undesirable for a CL method.** This is emphasized in CIFAR-10 simply because there are only 5 tasks (not because other datasets have more classes or fewer samples per class). For example if we re-run CIFAR-100 and MiniImagenet in the 5 task setting, we **observe that ER-ACE outperforms significantly SS-IL.**  Our method's final accuracy on the two datasets is **(26.6, 26.2)** vs **(23.8, 22.8)** for SS-IL.
> > > >
> > > > We hope that this addresses your concerns about the performance of our method. To summarize, we not only provide novel insight on the root causes of forgetting in online CL, but also propose a simple and state-of-the-art method based on this insight.
> > > >
> > > > Thank you.
> > > >
> > > > [†]  Aljundi et al. Online continual learning with maximally interfered retrieval, Neurips 2019.

---

### Official Review · Reviewer_qLmu · 2021-10-31

**Correctness:** 4
**Technical Novelty And Significance:** 2
**Empirical Novelty And Significance:** 4
**Recommendation:** 8
**Confidence:** 4

**Main Review:**

Strengths:
- The paper is clearly written and the main ideas are very well presented
- The motivation is clear and well illustrated by the experiments run in Figure 1
- Solid benchmarking effort. Many recent baselines were compared against, for a range of buffer sizes, and results reported with confidence intervals.

Weaknesses:
- It appears that the advantage of this method may start to break down when the data stream is a combination of new classes and old classes, since the old class representations would also start to be updated; is this the case? There are some preliminary experiments in Section 5.5, but I would have liked to see a much more thorough analysis - what happens when the number of examples per new class in the incoming data batch is very low? What about at a range of "blurriness" levels?


**Summary Of The Paper:**

This paper proposes a new update rule for the continual learning setting to boost performance when novel classes are added to the data stream.

**Summary Of The Review:**

Overall, the paper presents a strong analysis of a potential failure mode in continual learning, proposes a simple method as a fix, and compares thoroughly against baselines to validate their results.

---

> ### Author Response · Authors · 2021-11-12
> **Response to qLmu**
>
> Thank you so much for the kind words! We strongly believe that proper benchmarking is crucial, and truly appreciate this being recognized by reviewers. We have spent significant effort ensuring that our analysis (from baselines to hyperparameter budget to metrics reported) give a clear picture of real life desiderata for continual learning.
>
> (1) *Varying the blurriness of tasks.*
>
> Regarding your point on the relative impact of our method in settings with varying levels of task overlap. We observed that the impact of our method is significant whenever new classes are progressively introduced in the stream, irrespective of how “blurry” the tasks are. We took your suggestion and ran the following experiments to show this (all results averaged over 3 runs).  We use the CIFAR-10 benchmark with M=20, and use augmentations for a fair comparison across methods.
>
> We try different blurriness configurations, which control the overlap between tasks. To give an idea of how much the tasks overlap, we report the average number of unique classes per incoming minibatch : a small number means that the tasks are well separated. A high number means that there is a strong overlap. In the fully i.i.d setting, this number would be maximized. On the other hand, when this equals 1, each data class is streamed one after the other.
>
> | Method \ Inv. Bluriness | 1 | 2| 3| 4| 5|
> | ------------- | ----------- | -- |-- | -- | -- |
> | ER            | 23.1       |  25.7 |26.3 | 31.1| 34.4|
> | DER++     | 20.3       |  31.1 | 31.4 | 37.3| 36.7|
> | ER-ACE   | 32.8       |  36.2 | 36.8 | 41.7| 44.5|
> | ER-AML   | **34.0**       |  **40.4** | **46.0** |  **47.6**| **47.9**|
>
> We see that through a wide range of different blurriness levels, our methods show strong improvement over other task-free baselines
>
>
> (2) *Reducing the number of examples per task*
>
> Next, we evaluate the methods above using varying percentages of the training data from the second task onwards (we use all the data for the first task, so the model has converged to a reasonable solution before the first distribution shift). Moreover, we augment the rehearsal batch size for ER, ER-ACE and ER-AML to 20, so that their compute cost equals DER++.
>
>
> | Method \ % Samples used |5% | 10% | 25% | 50% |
> | ----------- | ----------- | -- |-- |-- |
> | ER      | 17.3 |22.5        | 28.0 | 33.2|
> | DER++   | 17.4 |19.9        | 24.8| 32.8|
> | ER-ACE  | **20.5** | **25.4**        | **31.2**| 36.1|
> | ER-AML  | 18.1 |24.3         |31.0| **38.7**|
>
> Again, we see that the proposed methods outperforms the baselines suggested above.
>
> Thank you.

---

> > ### Comment · Reviewer_qLmu · 2021-11-23
> > **Thanks for the additional experiments**
> >
> > Thank you for including the additional experiments. Indeed they make the paper more complete and present a more comprehensive view of algorithm performance under different conditions. It is interesting to note that ER-ACE outperforms ER-AML for a smaller sample size and vice versa for the blurriness experiments.

---

### Official Review · Reviewer_MwZR · 2021-11-02

**Correctness:** 3
**Technical Novelty And Significance:** 3
**Empirical Novelty And Significance:** 2
**Recommendation:** 6
**Confidence:** 4

**Main Review:**

This paper is well motivated.  Figure 1 is good to illustrate motivation.  It could be better to show the feature distribution with and without Asymmetric loss. Will the old class prototypes be pushed further away from the new ones?

The online continual learning scheme is valid and practical in real use cases. In an offline setting, after abundant iterations to replay old data, the performance can be recovered to some extent, but at the beginning stage, the accuracy will be drastically decreased.

Regarding the formulation of AML, the loss for buffer data is still in cross-entropy form, wondering if there is any special consideration not to design it in ML form by sampling positive and negative data.
AML is designed to be slightly more complex (sampling of positive and negative samples) yet less effective than ACE. It uses features from live training examples instead of the class weights. Could elaborate more on their respective resultant prototype distributions with AML and ACE.

From the experiment results, ER-AML leads to unstable performance while using data augmentation. The author can elaborate more on this.

More choices for the contractive loss could be explored. Also, the components of the proposed methods are not new, although it may be first used in this continual learning task.

Minor adjustment- N is used twice: "we denote the incoming N datapoints..." and "We use the P and N to denote the set of positive and negatives..."

**Summary Of The Paper:**

This paper studied one problem of continual learning which causes the forgetting of old classes, i.e new class representations will often overlap significantly with the previous classes, leading to highly disruptive parameter updates during the incremental training phase. To tackle this problem, this paper presented an Experience Replay (ER) based approach with asymmetric parameter update for old and new  data.  Two versions of asymmetric losses are proposed and compared: metric learning and cross-entropy based  losses. Experiments are conducted over multiple datasets to verify the effectiveness of the proposed method.

**Summary Of The Review:**

The overall idea of this paper is simple but effective. The motivation is well illustrated and the results are convincing. It can be improve with more insightful discussion and analysis on the outcome of the feature distribution (similar to Fig1) to show the proposed method can indeed separate the representation of old and new data along with the training.

---

> ### Author Response · Authors · 2021-11-12
> **Response to MwZR**
>
> Thank you for the feedback, and validating that Figure 1 illustrates the motivation behind our method! We will adjust the figure per your suggestions to make it even better.
>
> *Further Analysis of the Feature Distribution*
>
> We added a new analysis in Appendix I, showing the evolution of the representations and prototypes during learning of the second task (as in Figure 1). In summary, we show that
> (1) prototypes of Task 1 for ER as significantly displaced, which is not the case for ACE and AML.
> (2) ACE and AML converge to a solution where the prototypes align well with the representations from each class.
> (3) ACE converges more quickly than AML, which explains why it outperforms AML in settings with limited data per class.
>
> *On  why we use a cross-entropy loss for the rehearsal data in ER-AML*
>
> This is so that the model can perform efficient inference. If we instead use a metric loss for the rehearsal update, then we must resort to methods like Nearest-Class-Mean (NCM) to make evaluation predictions. NCM approaches (as in iCaRL) are very expensive in the online setting, as the class means need to be recomputed whenever the model weights are updated. In larger scale settings such as on the MiniImagenet with M=100, using a NCM procedure **increases the training compute by 17x** (from 118 to 2078 MFLOPS). This issue goes away when using a cross-entropy loss on the rehearsal data.
>
> *Further details on ER-AML*
>
> On the prototype distribution vs ER-ACE : we expect the alignment between the prototype and hidden representations from the same class to increase as more training data is presented to the learner. ACE’s prototypes are updated on all the streamed data, while AML’s prototypes are only updated on buffered data. Therefore, we see that in benchmarks with limited samples from each class (CIFAR-100 and Mini-Imagenet), ACE outperforms AML. However, when per class samples are abundant (therefore more per class samples end up in the buffer), AML obtains on-par or better performance (see CIFAR-10 results). To test this hypothesis, we artificially reduce the amount of training data for the second task onwards (same exp. as described in our answer to qLmu and in Appendix K). We see that ACE outperforms AML when limited data is available, however the trend shifts and the gap increases as more and more data is available to the learner.
>
> | Method \ % Samples used |5% | 10% | 25% | 50% | 100%|
> | ----------- | ----------- | -- |-- |-- | --|
> | ER-ACE  | **20.5** |**25.4**       | **31.2**| 36.1| 53.7 |
> | ER-AML  | 18.1 |24.3         |31.0| **38.7**|**55.7**|
>
>
>
> This suggests that ACE outperforms AML only in settings where limited data is available to train AML’s prototypes.
>
> *On trying other metric losses*
>
> In Appendix F and Table 5 we show results on CIFAR-10 when using the Triplet Loss. We observe a very similar trend : when (and only when) we carefully select negative samples to limit representation drift, we see significant improvements over baselines.
>
> Again, thank you for the precise feedback!

---

### Official Review · Reviewer_aUDf · 2021-11-03

**Correctness:** 3
**Technical Novelty And Significance:** 3
**Empirical Novelty And Significance:** 2
**Recommendation:** 5
**Confidence:** 2

**Main Review:**

The paper does good work explaining the phenomenon in the introduction section, which the reviewer found interesting. The paper also states the metric learning idea, which sounds plausible. Unfortunately, the writing somehow doesn’t connect this idea to the proposed method in section 4. The paper assumes the readers are familiar with the contrastive learning literature and doesn’t explain the loss functions quite well. It doesn’t explain why minimizing the loss function will lead to the effects mentioned in section 1. Since the reviewer is not from the area of representation learning, I couldn’t justify the correctness of the method. Besides this main issue, some other comments are as follows:
* The reviewer doesn’t find any experiments demonstrating the correctness of the method, i.e., whether the proposed method fixes the representation drift problem or not.
* The page format, i.e., margins between sections, were adjusted significantly and affects the reading.
* Add necessary parentheses over citations in Sections 3 and 4.
* Use a larger arrow in Fig. 1 (left) to represent gradients. It is hard to distinguish between gradients for tasks 1 and 2. The authors may want to explain how the gradient for task 1 is computed since the learner is in the task 2 phase.



**Summary Of The Paper:**

The paper proposed a new explanation of a performance drop phenomenon in a memory-based continual learning method, Experience Replay. The performance initially has a sharp drop when the task switch happens. As the training goes on, the performance drop can be improved but not as much as satisfactory. The paper analyzed this phenomenon from a representation learning perspective and applied a contrastive learning approach to reconciling the performance drop. The authors conducted experiments to illustrate their methods have a superior performance over baselines.

**Summary Of The Review:**

The paper analyzed an interesting representation drift problem of a popular continual learning method. However, the paper doesn’t explain the loss functions in detail. The reviewer couldn’t justify the correctness of the applied contrastive learning method.

---

> ### Author Response · Authors · 2021-11-12
> **Response to aUDf**
>
> Thank you for taking the time to review our paper! We will try to clarify several points which have caused confusion.
>
> *On the link between the issue raised in Fig. 1 and the loss functions proposed*
>
> In Figure 1, we highlight that the gradients from the new task back to the old task is disruptive (in orange). We address this issue by proposing 2 loss functions where this gradient is removed. For ER-AML (the metric loss approach) this gradient is not present. This is because for each incoming datapoint, we carefully select the positive and negative samples to belong to the current task : this avoids sending a disruptive gradient from the new task to the old task. Note that if we simply use the metric loss without this careful selection, the performance gains w.r.t to baselines go away (see Appendix G). For ER-ACE (the cross entropy approach), we can remove the disruptive gradient by ensuring the class prototypes from previous tasks are not updated in the incoming loss.
>
> *On the efficacy of our solution in solving representation drift*
>
> We break this down into two parts.
> First, does the method proposed limit abrupt changes in the representation ? Yes; we report this in Fig. 2 (Appendix H), where we see that our proposed loss functions have a significant impact on the representation drift.
> Second, does limiting the change in representation lead to better performance ? Yes; we see strong improvements over baselines which do not control for representation drift. In Fig. 1, we can also observe that our methods control much better for catastrophic forgetting.
>
> *On formatting and presentation issues*
>
> We have incorporated your comments and improved the spacing between sections and fixed missing citet
>
> Thank you!

---

### Author Response · Authors · 2021-11-12
**General Comments**

Dear Reviewers, we thank you for your valuable feedback! We really appreciate the time you spent providing us constructive advice to improve our submission. We have addressed your individual concerns in separate threads. Here we summarize the main changes in the submission pdf.

1. In-depth analysis showing the evolution during training of the representations and prototypes for ER, ER-ACE and ER-AML (Appendix I)


2. Additional experiments with blurry task boundaries (Appendix J)


3. New experiments varying the amount of training data available to the learner (Appendix K)

Thank you!

---

### Author Response · Authors · 2021-11-25
**Re: Discussion**

Dear Reviewers,

As the discussion period comes to an end, please let us know if you have any remaining questions. We have done our best to address all the concerns raised in your initial reviews, and are happy to engage with you, should you need further clarifications.

Thank you.

---

### Decision · Program_Chairs · 2022-01-20

**Decision:**

Accept (Poster)

**Comment:**

The manuscript develops new insights into how catastrophic forgetting takes place in the context of continual learning. The authors develop a new method based on this insight and demonstrate that it performs better than or as well as previously developed baselines, as well as showing that it is more widely applicable than close competitors (e.g. to cases where task boundary are unknown).
The manuscript starts by pointing to evidence that catastrophic forgetting at task boundaries is due at least in large part to abrupt representation drift (e.g. in the penultimate layer of a network) caused by gradients coming from new class examples. Most reviewers found this novel and interesting. Reviewers also tended to be happy with the writing, motivation, and experimental results supporting the conclusions.
One of the reviewers recommends against publishing (3 - Reject): mCT3 cites positives in the novel explanation of forgetting and the development of new metrics (Averaged Anytime Accuracy) and Total FLOPs, which they say help make the analysis more rigorous.
However, fundamentally, they believe that the relationship between the insights and the proposed methods are not strong enough and that the methods do not provide more than marginal improvements empirically. They point to work such as SS-IL as a baseline which, in their opinion, is not improved upon significantly enough to adjust their recommendation.
The authors provide multiple effective rebuttals to the concerns, as well as detailed experimental analysis of SS-IL: 0. The new method is shown to be as good or better than SS-IL, 1. That their method are more computationally and memory efficient than SS-IL, 2. SS-IL requires task-ids, whereas they do not, 3. Detail analysis (Appendix B) shows that SS-IL mostly fails to learn the current task in the online setting in the miniImageNet case that the reviewer worries about, a fact that is obscured by the simple Acc metric. Reviewer mCT3 does not respond to the rebuttals in any substantive or compelling fashion, and leaves their score at 3/Reject.
While I believe that the reviewers concerns should be thoroughly addressed in a final version of the manuscript, I am in agreement with the 3 of 4 reviewers who recommend publication.